# Structural and biochemical evidence supporting poly ADP-ribosylation in the bacterium *Deinococcus radiodurans*

Chao-Cheng Cho[1,2], Chia-Yu Chien[2], Yi-Chih Chiu[1], Meng-Hsuan Lin[1] & Chun-Hua Hsu [1,2]

Poly-ADP-ribosylation, a post-translational modification involved in various cellular processes, is well characterized in eukaryotes but thought to be devoid in bacteria. Here, we solve crystal structures of ADP-ribose–bound poly(ADP-ribose)glycohydrolase from the radioresistant bacterium *Deinococcus radiodurans* (DrPARG), revealing a solvent-accessible 2′-hydroxy group of ADP-ribose, which suggests that DrPARG may possess endo-glycohydrolase activity toward poly-ADP-ribose (PAR). We confirm the existence of PAR in *D. radiodurans* and show that disruption of DrPARG expression causes accumulation of endogenous PAR and compromises recovery from UV radiation damage. Moreover, endogenous PAR levels in *D. radiodurans* are elevated after UV irradiation, indicating that PAR-ylation may be involved in resistance to genotoxic stresses. These findings provide structural insights into a bacterial-type PARG and suggest the existence of a prokaryotic PARylation machinery that may be involved in stress responses.

---

[1] Genome and Systems Biology Degree Program, National Taiwan University and Academia Sinica, Taipei 10617, Taiwan. [2] Department of Agricultural Chemistry, National Taiwan University, Taipei 10617, Taiwan. These authors contributed equally: Chao-Cheng Cho, Chia-Yu Chien. Correspondence and requests for materials should be addressed to C.-H.H. (email: andyhsu@ntu.edu.tw)

P oly-ADP-ribosylation (PARylation) is a ubiquitous post-translational modification involved in various cellular processes, including gene expression, cellular signaling, protein activity and location, cell cycle progression, DNA repair, apoptosis, and necrosis[1–6]. In eukaryotes, PARylation has been widely studied for its relation to DNA damage because the formation of PAR polymers is regulated by three DNA damage-inducible poly (ADP-ribose) polymerases (PARPs): PARP1, PARP2, and PARP3[7–10]. In cells, the production of PAR polymers is mainly regulated by the activity of abundant nuclear PARP1.

Like other post-translational modifications that can be recognized by various cellular effectors, proteins containing specific domain structures such as macrodomains[11,12], PAR-binding linear motifs[13,14], PAR-binding zinc-fingers[15–17], and WWE domains[18] are associated with PAR polymers. To maintain normal cell functions, PAR-processing enzymes perform rapid turnover of PARylation to allow for temporal relocation of PAR-binding proteins as well as to form transient sub-organellar structures and signaling in the nucleus and cytoplasm[3].

Despite the abundance of PARP1, most of the PAR generated by DNA damage is rapidly degraded, with a half-life of less than 1 min by poly (ADP-ribose) glycohydrolase (PARG)[19,20]. Although some bacterial species are found to possess both PARP (closely related to PARP1[21]) and PARG genes and some contain only PARG homologs, bacteria are historically thought to lack poly(ADP-ribose) metabolism[22].

PARG has been the best-studied PAR-processing enzyme since its discovery in calf thymus extract four decades ago[23]. PARG can be divided into canonical and bacterial-type PARG based on domain structure composition. Vertebrates and most eukaryotes possess canonical PARGs that contain a complex N-terminal accessory domain followed by a catalytic domain[24–28]. Unlike canonical PARGs, bacterial-type PARGs found in some bacteria and a few eukaryotes (i.e., filamentous fungi, rotifer, and some protozoan species) have a non-conserved N-terminal extension[22,28]. A PARG-like macrodomain is found embedded in the catalytic domain of both canonical and bacterial PARGs[22,24–28].

The first PARG crystal structure derived from *Thermomonospora curvata* (TcPARG) revealed that the catalytic domain belongs to a distant member of the macrodomain protein family. The catalytic mechanism of TcPARG adopts a conserved glutamate residing in the PARG signature sequence to mediate nucleophilic attack of the putative oxocarbenium intermediate by a nearby water molecule, which results in the release of free ADP-ribose[22]. The TcPARG structure contains steric constraints resulting from the loop structure near its C terminus, called the ribose cap, which allows this bacterial-type PARG to bind to only the terminal residue of PAR polymers. Thus, the ribose cap structure confines bacterial-type PARGs to exo-glycohydrolases[22,28]. Structures of the canonical PARGs show that they share a highly similar mechanism of hydrolysis of PAR with bacterial-type PARGs[22,24–27], but possess both endo- and exo-glycohydrolase activities because they lack steric hinderance by the ribose cap structure. However, a conserved phenylalanine residue (Phe398 in *Tetrahymena thermophila* PARG; Phe902 in human PARG) accounting for low affinity binding of PAR in endo-mode caused canonical PARGs to predominantly act as exo-glycohydrolases[25,28,29]. As compared with several reported canonical PARG structures[24–27], only one bacterial PARG structure has been determined[22], which suggests still sparse knowledge of the diverse members of bacterial-type PARG family.

*Deinococcus radiodurans* and other members belonging to the same bacterial genus exhibit remarkable resistance to severe DNA damage caused by ionizing and ultraviolet (UV) radiation and many other agents that damage DNA[30]. *Deinococcus radiodurans*

has extraordinary ability to mend double-stranded DNA breaks (DSBs) following radiation damage. As compared with *Escherichia coli*, which mends only a few DSBs, *D. radiodurans* can mend more than 100 DSBs per chromosome within hours after irradiation[31,32].

The genome of *D. radiodurans* is completely sequenced, which facilitates extensive studies of mechanisms of radio-resistance at genomic and proteomic levels[33]. *Deinococcus radiodurans* possesses homologs of DNA repair machinery commonly adopted by other bacteria, such as UvrA and RecA for excision repair and homologous recombination, respectively[30,34–36]. Comparative genome analysis showed that *D. radiodurans* possesses fewer number of genes with known function involved in DNA repair pathways than other bacteria with larger genomes, such as *E. coli* and *Bacillus subtilis*, which suggests that the radio-resistant phenotype of *D. radiodurans* may result from unknown pathways as well as structural peculiarities of proteins that are not easily inferred from the sequences[37]. *Deinococcus radiodurans* is one of the many bacterial species possessing PARG homologs in their genomes[22,38]. Transcriptomic analysis showed upregulated expression of DrPARG following radiation damage, so it may be involved in DNA damage responses[39].

To determine the mechanistic basis of DrPARG-mediated ADP-ribose modification, we here solve crystal structures of DrPARG in the absence and presence of ADP-ribose. Structural comparison of DrPARG with previously characterized bacterial-type PARGs reveals differences in how they catalyzed PAR hydrolysis and implies a PAR metabolism in bacteria. Furthermore, we investigate DrPARG as a modulator of PAR formation in *D. radiodurans*. These findings may give insights into the function of bacterial PAR metabolism and add structural knowledge about diverse members of the bacterial PARG family.

## Results

**Structures of DrPARG in apo and ADP-ribose-bound forms.** The radio-resistant bacterium *D. radiodurans* encodes a PARG homolog, DrPARG, whose expression is upregulated after radiation damage, so it might play an important role in DNA repair[39]. Because the radio-resistance of *D. radiodurans* might result from subtle abnormalities of protein homologs of the conventional DNA repair machinery, which cannot easily be inferred from their primary sequences[37], we conducted structural and biochemical analyses of DrPARG to understand its activity at the molecular level. The crystal structure of DrPARG at 1.54 Å resolution reveals a typical macrodomain folding consisting of a seven-stranded β-sheet sandwiched between nine α-helices (Fig. 1a, Table 1). Eleven residues (GGGFLGGAQAQ) from position 100 to 110 in the apo-form structure of DrPARG, belonging to the catalytic loop of glycohydrolase, that were not well defined in the electron density maps were omitted. The crystal structure of the space group $P2_12_12_1$ of DrPARG contains one molecule per asymmetric unit. The inspection of lattice contacts indicates no formation of higher oligomers in the crystal, in accordance with gel filtration chromatography showing a $M_r$ of 32 kDa in solution (Supplementary Figure 1).

CD spectra of DrPARG in the absence and presence of ADP-ribose exhibited increased proportion of α-helices and β-strands from 11.78% and 30.37% to 15.10% and 33.52%, respectively, which suggests that conformational changes occurred during the binding of ADP-ribose (Supplementary Figure 2). Therefore, we attempted to obtain crystals of DrPARG in complex with ADP-ribose. Crystals were found in several similar conditions, but they belonged to different space groups. The monoclinic crystals of DrPARG in the ADP-ribose bound form gave the highest-resolution X-ray diffraction with space group $P2_1$ refined to 1.97

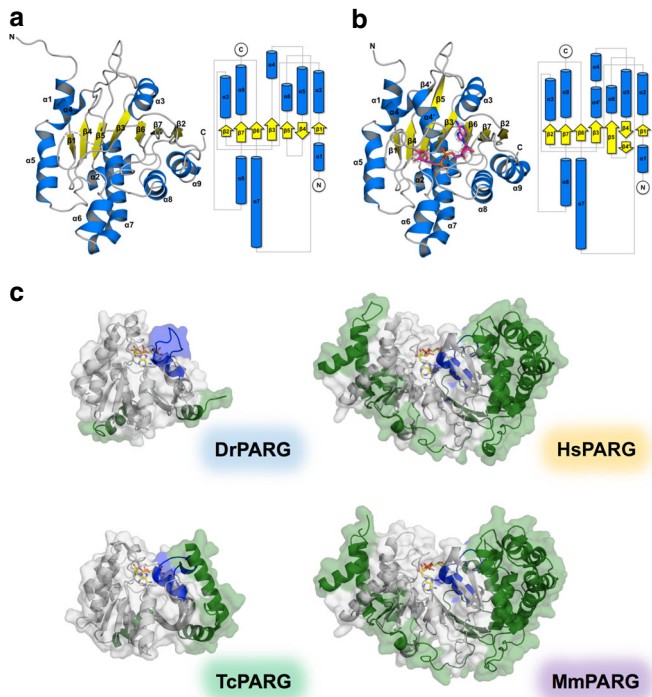

**Fig. 1** Crystal structures of apo and ADP-ribose-bound forms of DrPARG. **a**, **b** Structure and topology diagram of **a** apo and **b** ADP-ribose-bound forms of DrPARG. Left, structures are presented by cartoon models with helices, strands, and loops colored blue, yellow, and gray, respectively. The region not well defined in the electron density maps is omitted and shown as dashed lines. ADP-ribose is displayed in sticks with carbon in magenta, oxygen in red, nitrogen in blue, and phosphorous in orange. The $2F_o$-$F_c$ difference map, contoured at $1\sigma$, was calculated at 1.97 Å resolution from a model with the ligand omitted. Right, topology diagrams with the same colors as for cartoon representations. **c** PARG structures in complex with ADP-ribose from *Deinococcus radiodurans* (DrPARG; PDB code 5ZDB), *Thermomonospora curvata* (TcPARG; PDB code 3SIG), *Homo sapiens* (HsPARG; PDB code 4B1H), and *Mus musculus* (MmPARG; PDB code 4NA0) are shown as cartoon models. α4′ helix along with the catalytic loop of DrPARG and their corresponding regions of other PARGs are colored blue. The conserved macrodomain and the N- or C-terminal extension structures are colored gray and green, respectively. ADP-ribose molecules are shown as sticks with carbon, nitrogen, oxygen, and phosphorous colored yellow, blue, red and orange, respectively

Å resolution (Fig. 1b, Table 1). Notably, the 28 N-terminal residues of these solved structures from different space groups, including $P2_1$, $P3_2$, $P3_221$, and $P2_12_12_1$, were not well defined in the electron density maps, which suggests that this phenomenon did not result from crystal packing. In fact, the missing residues located in the intrinsically disordered N-terminal region of DrPARG (Supplementary Figure 3) and might be highly flexible.

**Unique features and conformational changes of DrPARG.** Upon ADP-ribose binding, as compared with the apo-form structure, that of DrPARG changes significantly (Fig. 1a). The missing 11 residues ($^{100}$GGGFLGGAQAQ$^{110}$) from the catalytic loop of the apo-form structure could be defined as a loop region in the ADP-ribose-bound form. As compared with the apo-form structure, the bound form structure of DrPARG showed additional secondary structures (Fig. 1b): a short β-strand (denoted β4′) following β4, extension of β5, and an α-helix (denoted α4′) preceding α4.

Thermal denaturation of DrPARG showed that the protein gradually unfolds below 40.0 °C with a calculated $T_m$ of 45.2 °C. In the presence of ADP-ribose, the denaturation rate decreased significantly below 40.0 °C along with increased $T_m$ by 46.5 °C, which indicates that binding of ADP-ribose contributes to the stabilization of DrPARG (Supplementary Figure 4). Consistent with the thermal denaturation study, structural analysis showed that in the bound form structure, the induced α4′ helix formed more intra-molecular hydrogen bonds, especially with residues from the loop connecting β3 and α4′ (Supplementary Figure 5A), whereas the equivalent region of the apo-form structure formed fewer hydrogen bonds with surrounding residues (Supplementary Figure 5B). Conformational changes of DrPARG might result from its highly flexible N terminus, which does not support the region of the α4′ helix and the catalytic loop during binding of ADP-ribose. As compared with DrPARG, previously reported bacterial and canonical PARG structures show corresponding regions of α4′ helix and the catalytic loop embedded between the conserved macrodomain and the N- or C-terminal extension structures (Fig. 1c). Thus, binding-induced conformational changes are less probable in these PARGs.

To gain insights into the molecular recognition of enzyme activity, we further investigated the binding pocket for ADP-ribose in the DrPARG structure and compared with those of representatives of canonical and bacterial PARG structures (Fig. 2). Like previously reported PARGs, coordination of ADP-ribose involves serial hydrogen bond formations and hydrophobic interactions provided by surrounding amino acid residues in DrPARG (Fig. 2a–c). The side chain of Asp113 located in the α4 helix contacts the N6 atom of the pyrimidine ring in the adenine moiety via direct hydrogen bonding. This residue is not well conserved among bacterial PARGs and may provide additional stabilization of ADP-ribose[22] (Supplementary Figure 6). Oxygen atoms of the pyrophosphate in ADP-ribose contact surrounding residues via hydrogen bonding with nitrogen atoms in the main chain of Gln110, Gly224, Gly226, Val227, and Phe228. The ribose″ moiety forms hydrogen bonds with Glu111, which is important for stabilizing the binding and catalytic activity in TcPARG[22]. Gly101 and Ser95 located near the ribose″ moiety, proposed to participate in binding of $(n-1)$ ADP-ribose moiety in the TcPARG-PAR model, are conserved in DrPARG[22]. In terms of generally similar binding mode of ADP-ribose among DrPARG, TcPARG, and PARG structures in complex with ADP-ribose from *Homo sapiens* (HsPARG) (Fig. 2a–c), the conserved catalytical residue Glu112 of DrPARG does not form a hydrogen bond with 1′-OH of the ribose″ moiety via its side chain (Fig. 2a and Supplementary Figure 6). Considering the flexibility of the catalytic loop in DrPARG, the side chain of Glu112 may re-orientate near the 1′-OH of ribose″ during actual catalysis. However, like in HsPARG, we observed a solvent-accessible 2′-OH of the ribose′ moiety in DrPARG (Fig. 2d–f), which suggests the internal binding of PAR polymers to DrPARG. The 2′-OH is structurally blocked in TcPARG and confines bacterial PARG to obligate exo-glycohydrolase[22,26] (Fig. 2e). In addition, lack of the extensional N-terminal helix of DrPARG, which restricts the motion of the α4′ helix along with the catalytic loop, may provide the entry site of PAR polymer from its internal part (Fig. 1c). Since our structural observation suggests a possible endo-glycohydrolase mode of DrPARG, we suspected that a PAR polymer might exist in the bacterium *D. radiodurans*.

**Endogenous PAR in *D. radiodurans*.** The discovery of protein homologs of the PARylation machinery suggests that functional PARylation exists in prokaryotic organisms[22,38]. However, characterization of the PARylation system has not been reported in

**Table 1 Data collection and refinement statistics**

| | Apo DrPARG | ADP-ribose-bound DrPARG | ADP-ribose-bound DrPARG | ADP-ribose-bound DrPARG | ADP-ribose-bound DrPARG | ADP-ribose-bound T267K | ADP-ribose-bound T267R |
|---|---|---|---|---|---|---|---|
| PDB code | 5ZDA | 5ZDB | 5ZDC | 5ZDD | 5ZDE | 5ZDF | 5ZDG |
| **Data collection** | | | | | | | |
| Space group | $P2_12_12_1$ | $P2_1$ | $P3_2$ | $P2_12_12_1$ | $P3_221$ | $P3_221$ | $P3_221$ |
| Cell dimensions | | | | | | | |
| $a, b, c$ (Å) | 45.1, 71.1, 72.3 | 43.3, 97.4, 59.4 | 108.5, 108.5, 131.0 | 42.7, 56.9, 97.5 | 62.9, 62.9, 131.3 | 63.2, 63.2, 130.9 | 63.0, 63.0, 131.0 |
| $\alpha, \beta, \gamma$ (°) | 90, 90, 90 | 90, 93.7, 90 | 90, 90, 120 | 90, 90, 90 | 90, 90, 120 | 90, 90, 120 | 90, 90, 120 |
| Resolution (Å) | 26.20-1.55 (1.61-1.55)[a] | 25.95-1.97 (2.04-1.97)[a] | 28.02-1.98 (2.05-1.98)[a] | 27.98-2.73 (2.85-2.73)[a] | 27.23-2.81 (2.91-2.81)[a] | 28.44-2.50 (2.59-2.50)[a] | 28.40-2.60 (2.69-2.60)[a] |
| $R_{merge}$ | 0.057 (0.487) | 0.086 (0.502) | 0.046 (0.512) | 0.117 (0.336) | 0.088 (0.507) | 0.147 (0.508) | 0.054 (0.579) |
| $I/\sigma I$ | 33.9 (4.3) | 14.3 (2.0) | 23.1 (2.4) | 13.6 (2.87) | 30.1 (5.1) | 15.4 (2.2) | 29.8 (2.6) |
| Completeness (%) | 97.8 (100.0) | 98.2 (91.1) | 99.1 (100.0) | 98.6 (90.7) | 99.9 (100.0) | 98.8 (90.4) | 99.9 (99.9) |
| Redundancy | 7.8 (8.1) | 3.8 (3.3) | 2.9 (2.8) | 5.2 (3.2) | 9.6 (8.9) | 6.9 (3.8) | 5.3 (5.3) |
| **Refinement** | | | | | | | |
| Resolution (Å) | 26.20-1.55 | 25.95-1.97 | 28.02-1.98 | 27.98-2.73 | 27.23-2.81 | 28.44-2.50 | 28.40-2.60 |
| No. of reflections | 33,710 | 32,430 | 112,158 | 6555 | 7429 | 9872 | 8899 |
| $R_{work}/R_{free}$ | 0.169/0.190 | 0.155/0.197 | 0.145/0.193 | 0.161/0.250 | 0.184/0.271 | 0.200/0.250 | 0.185/0.251 |
| No. of atoms | | | | | | | |
| Protein | 1910 | 3873 | 11,576 | 1916 | 1949 | 1915 | 1917 |
| Ligand/ion | 5 | 72 | 231 | 41 | 36 | 36 | 36 |
| Water | 285 | 266 | 1283 | 3 | — | 24 | 8 |
| B-factors | | | | | | | |
| Protein | 23.45 | 32.72 | 28.29 | 40.51 | 50.23 | 53.87 | 51.59 |
| Ligand/ion | 27.81 | 24.15 | 24.53 | 61.59 | 50.42 | 37.73 | 48.48 |
| Water | 35.06 | 39.46 | 37.81 | 37.66 | — | 38.10 | 42.31 |
| R.m.s. deviations | | | | | | | |
| Bond lengths (Å) | 0.006 | 0.008 | 0.008 | 0.010 | 0.009 | 0.009 | 0.009 |
| Bond angles (°) | 0.81 | 0.87 | 0.94 | 1.07 | 1.00 | 1.04 | 1.02 |

[a]Values within parentheses are for highest-resolution shell
*R.m.s.* root mean square, *DrPARG* PARG structure from *Deinococcus radiodurans*

any bacterial species. Although our structural analysis of DrPARG suggests that it can hydrolyze PAR, PARylation metabolism of *D. radiodurans* remains poorly understood. To determine whether endogenous PAR exists in *D. radiodurans*, we used a monoclonal antibody against PAR to detect the presence of inner PAR in the cell. As previously reported, the monoclonal PAR antibody could not detect signals from cell lysates of the budding yeast, considered to lack PARylation[22]. However, the PAR antibody showed strong signals from cell lysates of *D. radiodurans* (Fig. 3a). The PAR signal did not result from the cross-reactivity of antibody to nucleic acids in the cell lysates (Fig. 3b). The PAR signal increased significantly in lysates from *D. radiodurans* cultured in the TGY medium supplemented with additional NAD$^+$ (Fig. 3c). The detection result agreed with the feature of PAR formation utilizing NAD$^+$ as a substrate. Furthermore, NAD$^+$ or biotin-labeled NAD$^+$ was added to the culture of *D. radiodurans*. Incorporation of labeled NAD$^+$ into PAR was assayed by co-immunoprecipitation (Co-IP) with PAR antibody, thus suggesting that the PAR signal was dependent on the availability of NAD$^+$ (Fig. 3d). When the bacteria were treated with 3-aminobenzamide, a known inhibitor of ADP-ribosyltransferase, the PAR signal decreased (Fig. 3e), which was consistent with previously reported effect for decreased protein ADP-ribosylation in *Streptomyces* species[40,41]. The results suggested that endogenous PAR exists in *D. radiodurans*.

**Disruption of DrPARG causes endogenous PAR accumulation.**
To understand whether DrPARG functions in modulating endogenous PAR in *D. radiodurans*, we deleted the *parg* gene by

transformation of *D. radiodurans* with a plasmid containing the knock-out cassette composed of a *groEL* promoter-driven kanamycin-resistance gene flanked by genomic DNA segments 5′ and 3′ of the *parg* open reading frame (Supplementary Figure 7). After multiple rounds of selection for kanamycin resistance and the insertion of a drug resistance cassette, the absence of an intact *parg* gene was confirmed by PCR of genomic DNA with diagnostic primers (Supplementary Figure 6B). We monitored the growth of wild-type R1 and Δ*parg* strains under normal conditions and after UV irradiation. Under normal conditions at 30 °C, growth rate was slightly slower for Δ*parg* than R1 strain. However, Δ*parg* cells showed a significant difference in recovery rate after UV irradiation (Supplementary Figure 8). The role of DrPARG in DNA repair was further examined by pulsed-field gel electrophoresis (PFGE) and random-amplified polymorphic DNA (RAPD). The wild-type cells began the reassembly of genomes in 18–24 h after UV irradiation. In contrast to R1 strain, there was no evidence of genome reassembly in the Δ*parg* cells over this time course (Supplementary Figure 9). When the recovery from genotoxicity was also assayed by RAPD, the Δ*parg* strain exhibited a delayed restoration of genome integrity compared with the wild-type strain (Supplementary Figure 10). In non-irradiated R1 cells, the PCR generated two major bands between 1.5 and 2.5 kb. After UV irradiation, the intensities of the two major bands decreased. During recovery, additional bands appeared between 1.0 and 1.5 kb. The major bands restored the intensities and patterns to the non-irradiated states in 12–18 h, indicating that R1 strain began to regain its genome integrity. In non-irradiated Δ*parg* cells, the PCR generated three major bands

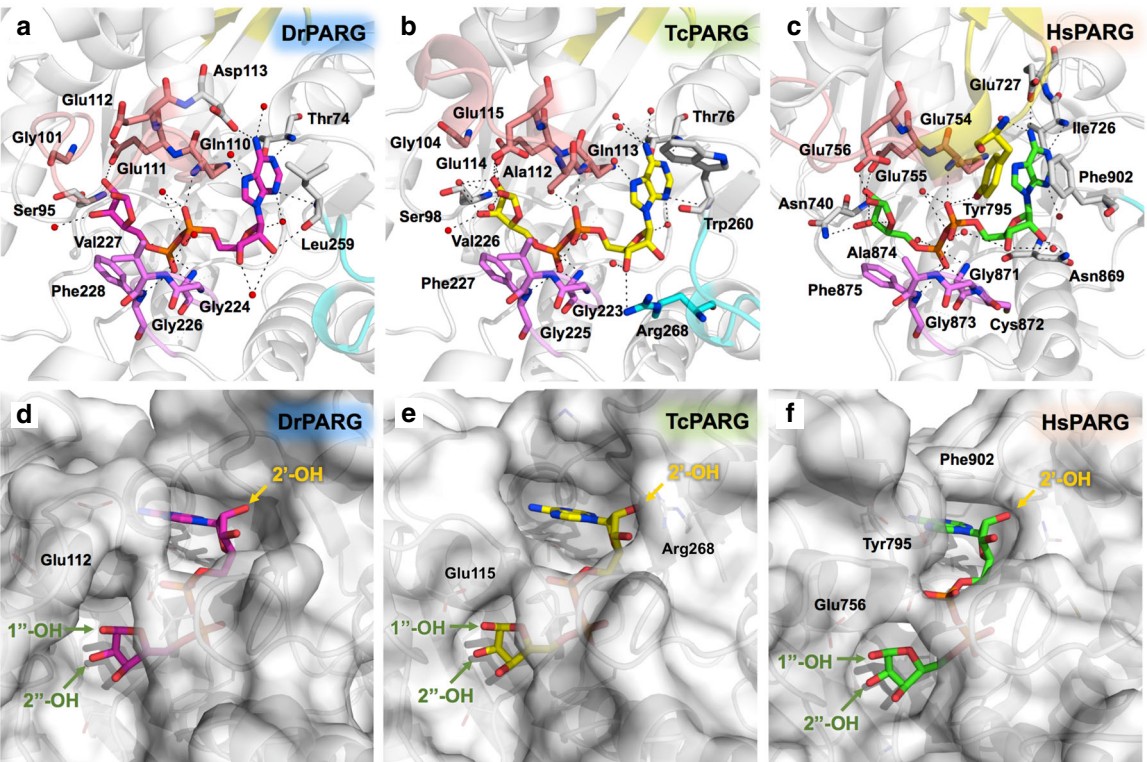

**Fig. 2** Detailed view of the ADP-ribose binding pocket of DrPARG. **A** detailed comparison of interactions within ADP-ribose binding pocket of PARG structures from **a** *Deinococcus radiodurans* (DrPARG), **b** *Thermomonospora curvata* (TcPARG), and **c** *Homo sapiens* (HsPARG). Proteins are represented as cartoon diagram with key residues and ADP-ribose shown as sticks. Carbons of ADP-ribose are colored individually among the three structures. Carbons of the catalytic loop, diphosphate-binding loop, tyrosine clasp, and ribose cap are colored salmon, purple, yellow, and cyan, respectively. Oxygen, nitrogen, and phosphorus are colored red, blue, and orange, respectively. Crystallized waters are shown as red spheres. Hydrogen bonds are represented as dashed lines. Solvent accessibility of bound ADP-ribose in **d** DrPARG, **e** TcPARG, and **f** HsPARG. ADP-ribose is shown as sticks with carbons in the same colors as with panel **a**, **b**, and **c**. Selected residues are labeled. Ribose hydroxyls with potential poly-ADP-ribose (PAR) linkage points are labeled and indicated by arrowheads

between 0.75 to 2.5 kb; however, unlike R1 strain, the major bands disappeared after irradiation and showed no sign of restoration until 18 h. The results suggested that DrPARG may involve in DNA damage repair in *D. radiodurans*. We further detected PAR signals in R1 and Δ*parg* cell lysates resolved in acrylamide gel by blotting with monoclonal PAR antibody or pan-ADP-ribose-binding reagent[41] (Fig. 3f). The two reagents generated different profiles. This might result from their differential affinities to the structural factors of bacterial PARylation such as length and branch. Nevertheless, some protein bands showed higher intensities in Δ*parg* cells compared with those in R1 cells detected by both reagents (Fig. 3f), suggesting the modification of protein was affected by PARG status. Endogenous PAR level showed an overall increase by ~2-fold in Δ*parg* cells compared with that in R1 cells (Fig. 3g). The results suggested DrPARG involves in the regulation of PARylation in *D. radiodurans*. When the lysates of Δ*parg* cells were treated with HsPARG, PAR level decreased (Fig. 3h). This indicated endogenous PAR of bacteria can be recognized and processed by canonical PARG and may structurally resemble to that of higher eukaryotes.

Previous transcriptomic analysis of gene expression profiles after radiation damage showed immediate elevation of DrPARG messenger RNA level within 1.5 h after application of ionizing radiation[39]. To verify the role of DrPARG in resistance of *D. radiodurans* to radiation damage, we monitored the PARylation level in R1 and Δ*parg* cells after UV irradiation (Supplementary Figure 11). In the early recovery phase (0–3 h), PAR level in R1 cells peaked at 1 h after irradiation and gradually decreased

after 1 h (Supplementary Figure 11A), but in Δ*parg* cells during the early recovery phase, PAR level accumulated up to 3 h and gradually decreased in the mid recovery phase (3–9 h) (Supplementary Figure 11B). Interestingly, the dynamics of PAR level in R1 cells agreed with DrPARG expression induced immediately after radiation damage[39]. Thus, PARylation might be involved in the immediate response to radiation damage in *D. radiodurans*. The decrease in PAR level in Δ*parg* cells during the mid-recovery phase might result from induced expression of other PAR-processing enzymes, such as nucleoside diphosphate-linked motif X (Nudix) family hydrolases[42] or ADP-ribosyl-glycohydrolase (ARH3) family proteins[43].

**DrPARG acts in both exo- and endo-glycohydrolase modes**. We observed a 2′-OH ribose′ moiety exposed to solvent in the ADP-ribose-bound structure of DrPARG, which suggests that an extra $n + 1$ ADP-ribose unit could fit in the protein surface. The finding prompted us to speculate that DrPARG might possess endo-glycohydrolase activity even though bacterial PARG has been reported to act in solely exo-glycohydrolase mode[26]. To test this hypothesis, we constructed the structural model of DrPARG in complex with tri-ADP-ribose using molecular dynamics (MD) simulation (Fig. 4a) and evaluated it by comparing the modeled PAR with structures of the obligate exo-glycohydorlase TcPARG and the observed ADP-ribose-bound DrPARG. As compared with the TcPARG structure, in the ADP-ribose-bound DrPARG, the ribose″ moiety of $n + 1$ showed a steric clash with proximal Arg268, with the side-chain position strictly coordinated by

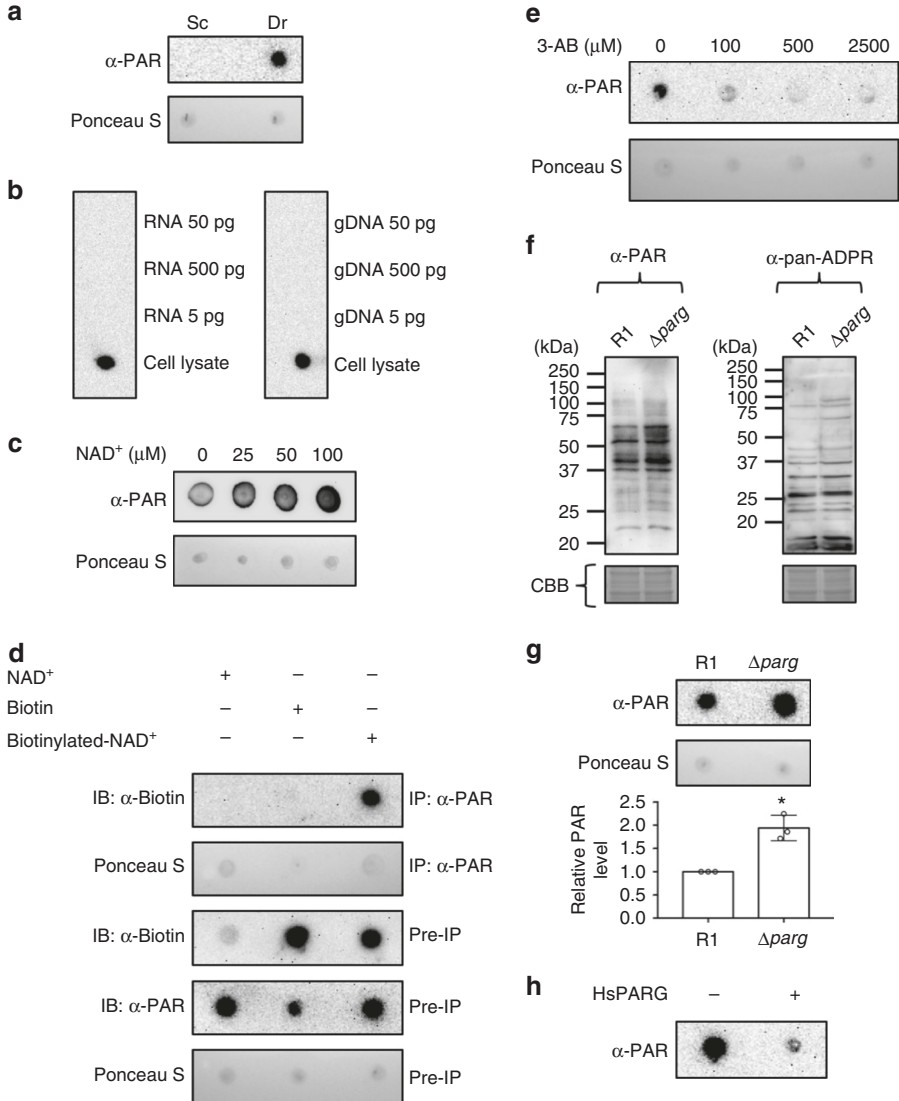

**Fig. 3** Detection of endogenous poly-ADP-ribose (PAR) in *Deinococcus radiodurans*. **a** PAR formation assayed in cell lysates of *D. radiodurans* R1 (Dr) by dot blotting. The cell lysates of *Saccharomyces cerevisiae* BY4741 (Sc) were used as a negative control. The membrane was stained with Ponceau Red as a loading control. **b** Various amounts of genomic DNA and RNA of *D. radiodurans* R1 were spotted on nitrocellulose membrane and assayed for PAR signal by dot blotting. The cell lysates of R1 were used as a positive control. **c** PAR levels assayed by dot blotting in *D. radiodurans* R1 cultured with $NAD^+$ supplement. Ponceau Red staining is the loading control. **d** The incorporation of $NAD^+$ into endogenous PAR in *D. radiodurans* was assayed by co-immunoprecipitation (IP) and immunoblotting (IB). Biotin-labeled or -non-labeled $NAD^+$ was supplemented to R1 culture. Pre-IP or IPed samples were analyzed by indicated antibodies or Ponceau Red staining. R1 culture supplemented with biotin was used as a control for the entry of biotin-labeled $NAD^+$ into R1 cells. **e** PAR levels were assayed by dot blotting in *D. radiodurans* R1 culture treated with various amounts of 3-aminobenzamide (3-AB). Ponceau Red staining is the loading control. **f** The cell lysates of *D. radiodurans* R1 and $\Delta parg$ strains were resolved in 12% acrylamide gel and blotted with PAR antibody and anti-pan-ADP-ribose binding reagent. The gels were stained with Coomassie Brilliant Blue (CBB) as a loading control. **g** PAR formation was assayed in wild-type *D. radiodurans* R1 and $\Delta parg$ strains cultured in the TGY medium at 30 °C by dot blotting. The membrane was stained with Ponceau Red as a loading control. PAR levels were quantified by using ImageJ and plotted as mean ± SEM ($n = 3$ independent experiments) relative to the R1 strain, with PAR levels set to 1. *$P \leq 0.05$, determined by Student's *t*-test. **h** The cell lysates of $\Delta parg$ strain were incubated with or without recombinant human PARG (PARG from *Homo sapiens* (HsPARG)) and PAR levels were assayed by dot blotting. Source data are provided as a Source Data file

Asp261, Cys224, and ribose′ of the *n* unit ADP-ribose via hydrogen bonding[26] (Supplementary Figure 12). Besides avoiding a direct steric clash with ionic pairing formed by side chains of Arg268 and Asp261, the binding of PAR in endo-mode requires reorientation of the adenine moiety backward to near the α6 helix, which is also unfavorable for accommodating the *n* unit ADP-ribose by creating steric hindrance in the adenine cavity (Fig. 4b). However, the modeled tri-ADP-ribose fit ideally in the ADP-ribose-bound DrPARG structure, which was considered to be in its endo-glycohydorlase mode. The barrier created by ionic

paring as well as hydrogen bonding of Arg268 and Asp261 in TcPARG was not observed in the DrPARG structure. A substitution of Arg268 for the smaller and uncharged side chain of Thr267 than that of arginine disrupted the interaction matrix of Thr267, Asp260, and *n* ribose′ and allowed for accommodating the *n* + 1 ADP-ribose (Fig. 4c and Supplementary Figure 12). The inducible α4′ helix of DrPARG might also contribute to PAR binding by changing its conformation to accommodate the adenine moiety of the *n* unit ADP-ribose (Fig. 4c). The observation

suggests the existence of both exo- and endo-glycohydrolase modes in DrPARG.

**Structure-guided mutagenesis of DrPARG.** To clarify the effect of Thr267 on enzyme activity of DrPARG, we generated mutants of DrPARG by substituting Thr267 for arginine or lysine. PARG cleavage assay in vitro showed T267R and T267K mutants with decreased hydrolysis PAR activity as compared with the wild-type enzyme. Substituting Glu112 for alanine, considered for catalytical inactivation, abolished the PAR hydrolysis activity of DrPARG (Supplementary Figure 13). Circular dichroism (CD) spectroscopy used to analyze wild-type and mutant DrPARG for folding and thermal denaturation showed no significant differences in folding and $T_m$ between the groups (Supplementary Figure 14), which suggests that differential PAR hydrolysis activities did not result from mutation-causing structural abnormality.

To precisely compare the PAR hydrolyzing activity of DrPARG mutants with the wild-type, we examined the kinetic parameters from enzyme kinetics (Table 2, representative Michaelis–Menten plots are shown in Fig. 5a). Consistent with in vitro cleavage assay, the mutants designed for blocking endo-glycohydrolase activity showed significantly decreased turnover rate Kcat as compared with the wild-type DrPARG, so the mutants may have reduced activity toward PAR. The mutants also showed increased Michaelis constant $K_m$, which indicates decreased binding affinity of PAR. Detection of PAR in the cleavage products of DrPARG was examined by using a 30 kDa cut-off filter to separate short chains of PAR from unhydrolyzed PAR attached to protein in the in vitro cleavage assay (Fig. 5b). Short-chain PAR began to appear in the filtrate of DrPARG at 5 min post treatment, peaked at 10 min, and further decreased at 15 min. The same results were also observed for HsPARG. Thus, DrPARG could act in endo-mode to produce short-chain PAR, which was subsequently hydrolyzed to smaller oligo ADP-ribose beyond the detection limit of PAR antibody[44]. Short-chain PAR was not detected in the filtrates of the two DrPARG mutants T267R and T267K, as well as the TcPARG, although the mutants still hydrolyzed PAR (Fig. 5a, Table 2, Supplementary Figure 13). The detected short-chain PAR did not likely result from removal of short PAR from PARP1, but rather from endo-cleavage of O-glycosidic linkages of long-chain PAR because DrPARG could not hydrolyze glutamate- or serine-

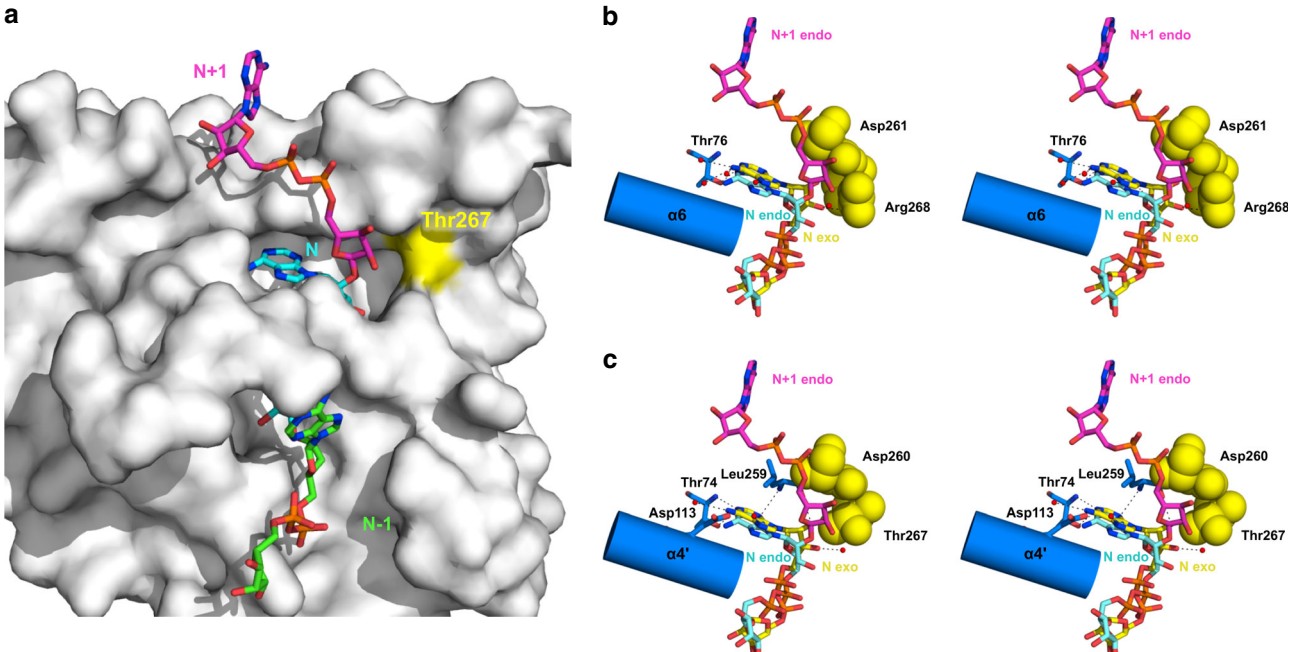

**Fig. 4** Structural model of DrPARG-PAR (poly-ADP-ribose) complex in endo-glycohydrolase mode. **a** Solvent-accessible surface of DrPARG in gray (position of Thr267 is in yellow) with a PAR₃ modeled in an endo-glycohydrolase position. Carbons of the three individual ADP-ribose units are distinctly colored. **b**, **c** Stereo images of an overlay of the modeled endo-glycohydrolase PAR₃ (colored the same as in **a**) with the **b** obligate exo-glycohydrolase PARG structures in complex with ADP-ribose from *Thermomonospora curvata* (TcPARG) structure and **c** observed exo-glycohydrolase DrPARG structure. Residues and secondary structures implicated in steric hindrance with n + 1 and n ADP-ribose units of TcPARG as well as their equivalents of DrPARG are shown as spheres (colored yellow) and a cartoon model, respectively. The hydrogen network between the adenine moiety/ribose′ of the n ADP-ribose unit and the protein/crystallized water observed for the exo-glycohydrolase mode is shown as dashed lines

**Table 2 Kinetic parameters of enzymatic hydrolysis of PAR**

| Protein | $K_m(\mu M)$ | $V_{max}$ (μmol/min/mg protein) | $k_{cat}$ (1/s) | $k_{cat}/K_m$ (1/s/μM) |
|---|---|---|---|---|
| WT | 2.96 ± 0.90 | 515.60 ± 62.90 | 266.40 ± 32.50 | 90.00 ± 29.47 |
| T267R | 9.34 ± 1.92 | 384.30 ± 46.81 | 198.60 ± 24.19 | 21.26 ± 4.92 |
| T267K | 11.14 ± 2.75 | 351.00 ± 53.73 | 181.40 ± 27.76 | 16.28 ± 4.73 |
| E112A | ND | ND | ND | ND |

Values are mean plus or minus SEM (n = 3 independent experiments). Fitted Michaelis–Menten plots are shown in Fig. 5a
*PAR* poly-ADP-ribose, *WT* wild type, *ND* no enzymatic activity detected

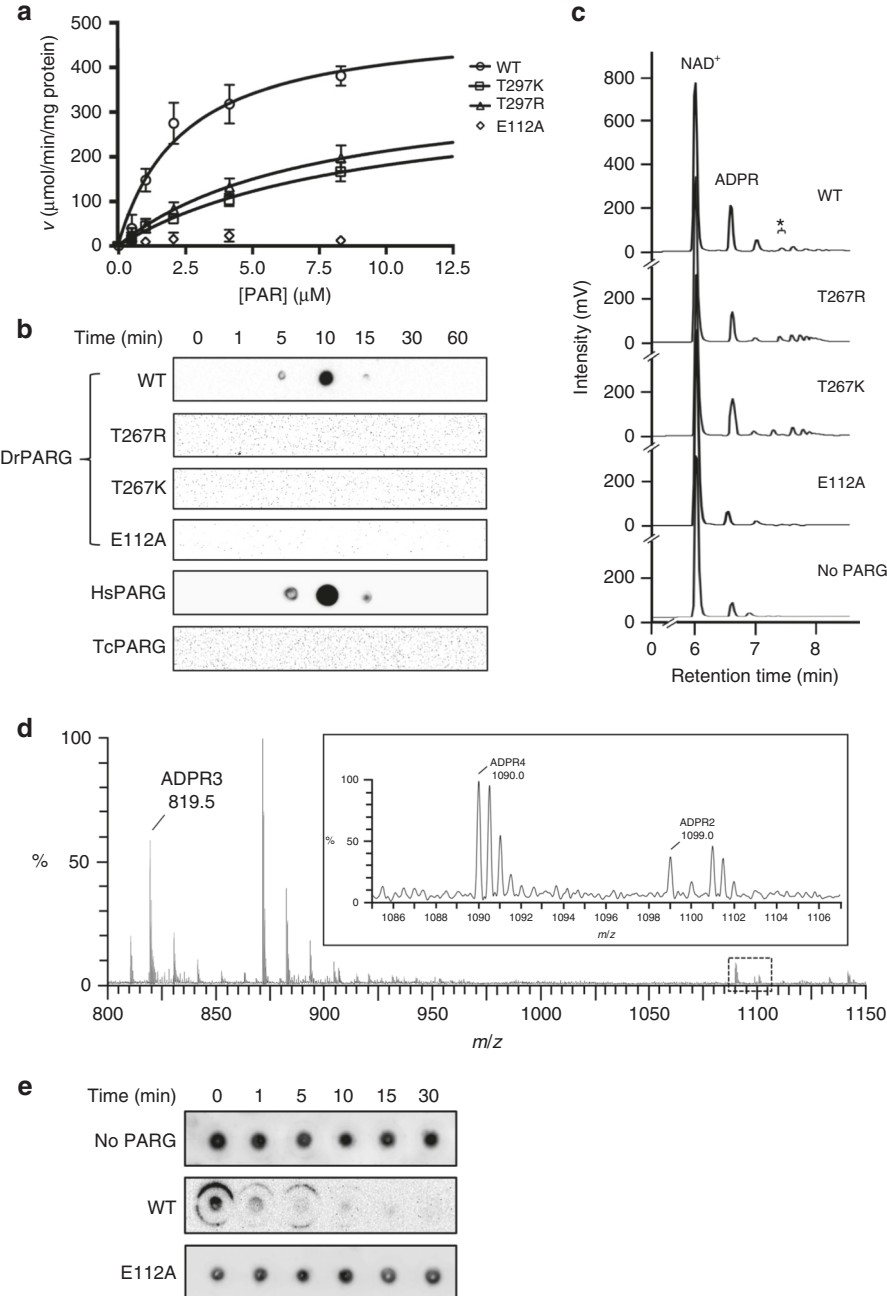

**Fig. 5** Enzyme activity of DrPARG. **a** Michaelis–Menten plots of wild-type (WT), T267R, T267K, and E112A DrPARG catalyzing poly-ADP-ribose (PAR) hydrolysis are shown. Initial reaction velocity (μmol/min/mg protein) is plotted against PAR substrate concentration (μM). Data are plotted as mean ± SEM ($n = 3$ independent experiments). WT, T267R, T267K, and E112A DrPARG are indicated by open circle, triangle, square, and diamond, respectively. **b** Detection of PAR in the cleavage products of DrPARG. The reaction of automodified PARP1 treated with or without (time = 0) WT and mutant DrPARG was stopped at different times and filtered with a 30-kDa cut-off column. PAR was assayed in the filtrate by dot blotting. PARGs from *Homo sapiens* (HsPARG) and *Thermomonospora curvata* (TcPARG) were used as controls of canonical and bacterial PARG, respectively. **c** The products of reactions treated with WT and mutant DrPARG for 15 min were filtered with a 10-kDa cut-off column and subjected to high-performance liquid chromatography (HPLC) analyses. ADP-ribose (ADPR) and residual $NAD^+$ were indicated. The fraction indicated by an asterisk from cleavage products of WT DrPARG was collected and analyzed by mass spectrometry. **d** Identification of endo-glycohydrolase cleavage products of DrPARG by time-of-flight mass spectrometry (TOF MS). The cleavage products separated by HPLC were subjected to a quadrupole time-of-flight (Q-TOF) instrument for analysis. Ion chromatogram was shown with *m/z* ratios for deprotonated ADPR dimer (ADPR2; *m/z* 1099.0), doubly charged deprotonated ADPR trimer (ADPR3; *m/z* 819.5), and doubly charged deprotonated ADPR tetramer (ADPR4; *m/z* 1090.0) indicated. **e** Enzyme activity of DrPARG toward endogenous PAR of *D. radiodurans* was assayed in cell lysates from the Δ*parg* strain treated with or without (time = 0) WT or E122A DrPARG. PAR level was assayed by dot blotting over a time course. Source data are provided as a Source Data file

linked mono ADP-ribosylation[41,45] (Supplementary Figure 15). When the cleavage products of wild-type DrPARG were separated by high-performance liquid chromatography (HPLC) and further analyzed by mass spectrometry (MS), oligo ADP-ribose such as dimer, trimer, and tetramer were identified (Fig. 5c, d). The results suggested that DrPARG possesses endo-glycohydrolase activity. The two DrPARG mutants T267R and T267K, however, still produce oligo ADP-ribose (Fig. 5c). Thus, such substitution may compromise but not abolish the endo-glycohydrolase activity of DrPARG. In other words, Thr267 is an important but not essential factor to endo-glycohydrolase activity of DrPARG. The observations supported the structurally flexible regions that contribute to establishing endo-glycohydolase activity (Fig. 4c).

To further compare the ability of DrPARG mutants to bind ADP-ribose with the wild-type DrPARG in exo-mode, the thermodynamics of the wild-type or mutant DrPARG association with ADP-ribose were compared by using isothermal titration calorimetry (ITC; Table 3, representative binding isotherms are shown in Supplementary Figure 16). The result suggested that the mutation (T267R, T267K, or E112A) had little or no impact on affinity to ADP-ribose; however, substitution of Glu112 with alanine conferred unfavorable enthalpy change, which might result from the mutation somehow affecting hydrogen bonding between ADP-ribose and its adjacent residues. Comparison of the affinity of wild-type and mutated DrPARGs to bind ADP-ribose indicated that mutation on Thr267 did not affect the affinity to bind the external ADP-ribose unit.

Furthermore, we determined crystal structures of DrPARG T267R and T267K in complex with ADP-ribose for understanding the detailed interactions (Table 1, Supplementary Figure 17). The crystal structures of T267R and T267K belonged to the space group $P3_221$. In consistent to the secondary structure analyzed by CD spectrum (Supplementary Figure 14A), the substitution of the residue did not cause dramatic alterations to overall structure compared with the wild-type enzyme with root mean square deviation to wild type of 0.50 and 0.43 Å for T267R and T267K, respectively. As compared with the ADP-ribose bound TcPARG structure, the side-chain position of Arg267 was fixed like that of TcPARG by regaining complex hydrogen bonding with ADP-ribose and neighboring His262 and Cys225 as well as ionic interaction with Asp260 (Supplementary Figure 17A), which suggested that the endo-glycohydrolase activity was impaired. The side chain of Lys267 did not form hydrogen bonding with neighboring residues, but positioned toward the side chain of Asp260 by ionic interaction (Supplementary Figure 17B). The superposition of the modeled PAR trimer and the DrPARG T267K structure showed that the $n + 1$ unit was blocked by the barrier resulted from ionic pairing of side chains of Lys267 and Asp260 and implicated that the ionic bonding might be sufficient for establishing the steric hindrance of the exo-glycohydrolase mode (Supplementary Figure 17C). Compared with the solvent-accessible 2′-OH with $n + 1$ linkage point observed in wild-type (Supplementary Figure 17D), which were

inaccessible and less accessible to solvent in the complex structures of T267R and T267K, respectively (Supplementary Figure 17E, 17F). Since T267R and T267K DrPARG catalyze the hydrolysis of PAR without generating short-chain PAR but oligo ADP-ribose (Fig. 5a–c), the establishment of endo-glycohydrolase activity was not solely accounted from the lack of steric hinderance near 2′-OH. The flexibility of the binding pocket for ADP-ribose might contribute to endo-glycohydrolase activity by accommodating PAR during catalysis.

**DrPARG processes endogenous PAR of *D. radiodurans*.** Our biological data showed that disruption of DrPARG caused the accumulation of endogenous PAR in *D. radiodurans* (Fig. 3g). To determine whether DrPARG directly processes endogenous PAR of *D. radiodurans*, we used in vitro activity assays of wild-type DrPARG and the E112A inactive mutant to process endogenous PAR in Δ*parg* cell lysates. In vitro cleavage assay revealed that cellular PAR was digested over the time course by incubation with the wild-type DrPARG but not DrPARG E112A mutant (Fig. 5e), so DrPARG possesses direct activity for processing the endogenous PAR of *D. radiodurans* and further implies that DrPARG might play an important role in the PARylation metabolism of *D. radiodurans*.

## Discussion
Very little is known about the existence and function of bacterial PARylation. However, increasing studies of the discovery of bacterial homologs of the key enzymes involved in PARylation metabolism such as PARP, PARG, and TARG have suggested that functional PARylation systems may exist in prokaryotes[22,38,46]. We provide a reliable and reproducible detection of endogenous PAR in *D. radiodurans* with the PAR monoclonal antibody. The PAR antibody has broad applications in detecting PAR from mammal as well as plant and protozoan cells[47–49], thereby suggesting that PAR produced by divergent organisms is structurally alike and can be recognized by specific antibodies. However, the genome of *D. radiodurans* does not appear to encode PARP homologs. Mono ADP-ribosyltransferase (ART), such as mammalian ART2, can generate PAR polymers, so the activity of producing PAR may not be restricted to PARP family members[50]. Mono ADP-ribosylation is characterized more in bacteria because their genomes encode ADP-ribosylating toxins (such as cholera toxin and diphtheria toxin) and sirtuins[51,52]. The PARrylation machinery seems probable in *D. radiodurans* because its genome encodes a sirtuin homolog. When *D. radiodurans* was treated with 3-aminobenzamide, a known inhibitor of ADP-ribosyl-transferase, endogenous PAR decreased. Thus, *D. radiodurans* may generate PAR by a highly divergent PARP that awaits to be uncovered or by the enzymes with known ADP-ribosylating function acting as alternative PARPs.

Our study demonstrates that DrPARG is involved in catalyzing the breakdown of endogenous PAR and DNA repair in *D. radiodurans*. However, the connection between PAR metabolism and DNA damage repair is still unclear. Recent studies have

**Table 3 Summary of thermodynamic parameters for ADP-ribose binding**

| Protein | $K_d$ (µM) | n | $\Delta H$ (kJ/mol) | $-T\Delta S$ (kJ/mol) | $\Delta G$ (kJ/mol) |
|---------|-----------|---|-----------|------------|-----------|
| WT | 1.78 ± 0.05 | 1.05 ± 0.05 | −89.45 ± 3.08 | 56.64 ± 3.06 | −32.80 ± 0.08 |
| T267R | 3.76 ± 0.47 | 1.09 ± 0.01 | −90.44 ± 2.96 | 59.47 ± 3.26 | −30.96 ± 0.30 |
| T267K | 1.98 ± 0.02 | 1.05 ± 0.03 | −76.38 ± 0.97 | 43.84 ± 0.97 | −32.53 ± 0.02 |
| E112A | 2.02 ± 0.07 | 1.06 ± 0.04 | −40.38 ± 1.59 | 7.88 ± 1.50 | −32.50 ± 0.09 |

All experiments were carried out at 25 °C. Values are mean plus or minus SEM (n = 3 independent experiments). Plots of raw data and fitted curve are in Supplementary Fig. S16
WT wild type

demonstrated that ADP-ribosylation of serine is the primary modification after DNA damage and its turnover is dependent on ARH3[41,53,54]. Since possessing both PARG and ARH3 homologs, *D. radiodurans* may be capable of fully reversing ADP-ribosylation of proteins. Notably, the level of endogenous PAR greatly increases during the early phase (0–3 h) of recovery after radiation damage. Given that the expression of DrPARG also increases immediately after irradiation, PARylation may be involved in the DNA damage response in *D. radiodurans*. During the early recovery phase, cell growth is inhibited and *recA* is induced[39]. Whether PARylation occurs in certain residues of the components belonging to DNA repair machinery in *D. radiodurans* needs further investigation.

Our structural and biochemical analyses of DrPARG show that its N terminus and catalytic loop are highly flexible. The structural flexibility may reflect the multiple space groups observed in X-ray crystallography because packing patterns are constantly influenced by the flexible regions during crystallization. The presence of inducible secondary structures observed from the comparison of apo- and bound-form structures confirms that conformational changes occur during ADP-ribose binding to DrPARG. The previously reported crystal structure of TcPARG suggests that the N terminus supports the catalytic loop and confines the TcPARG structure in a relatively rigid state as compared with DrPARG[22]. TcPARG is an obligate exo-glycohydrolase owing to the steric hindrance created by the side chain of Arg268, which is coordinated by complex hydrogen bonding with Asp261 and neighboring residues as well as ADP-ribose[26]. In DrPARG, the corresponding residue to Arg268 of TcPARG is Thr267, for which the side chain is uncharged as compared with arginine, does not form ionic and hydrogen bonding with Asp260 and neighboring residues, and makes room for accommodating the $n + 1$ ADP-ribose unit in endo-glycohydrolase mode. Binding of the $n + 1$ unit also causes reorientation of the adenine moiety of the $n$ ADP-ribose unit toward an inducible α helix, which may contribute to endo-mode PAR binding by changing its conformation. Our mutational analyses suggest that besides the lack of steric hinderance at Thr267, structural flexibility may contribute to endo-glycohydrolase activity of DrPARG. Our data suggest the presence of both exo- and endo-activities of PARG in prokaryotic microorganisms. Potential bacterial PARGs can be identified by BLAST searches (Supplementary Table 1). Although Thr267 is less conserved among these proteins than Asp260 (Supplementary Figure 18), many of them have threonine or valine in the position corresponding to Thr267, which may not form ionic pairing with aspartate. However, the number of bacterial endo-PARG may be underestimated since Thr267 is not the only determinant of endo-glycohydrolase activity.

Bacteria may acquire genes encoding PARGs via horizontal gene transfer[22,38]. Given that the sequences of PARG in *T. curvata* and *D. radiodurans* are similar (sequence identity 49%), TcPARG and DrPARG may have a common origin, but separately evolved by introducing unique structural features to help different bacterial species adopt individual growing environments. Our data show that DrPARG directly processes the cellular PAR of *D. radiodurans*. For *D. radiodurans*, a PARG equipped with both exo- and endo-glycohydrolase activities may efficiently mediate cellular PARylation to survive the genotoxic stress. *Deinococcus radiodurans* is one of the representatives of an ancient group with no clear affinities to any other known bacterial lineages[37]. Our study indicates that PAR metabolism is conserved in primitive prokaryotes and its connections with DNA damage response may be more ancient than previously considered.

Taken together, our findings provide evidence of poly-ADP-ribosylation and its degradation in *D. radiodurans* belonging to an ancient lineage of bacteria. The study may help understand poly-ADP-ribosylation and its potential connection to the DNA damage response in *D. radiodurans* and establish *D. radiodurans* as a model organism to study the bacterial poly-ADP-ribosylation system. From observing the structural peculiarities of DrPARG, we show that although previously thought to be an obligate exo-glycohydrolase, bacterial PARG could exert both exo- and endo-glycohydrolase functions. We believe our findings may pave the way for further investigations of the regulation and function of poly-ADP-ribosylation in prokaryotic organisms.

## Methods

**Protein expression and purification**. The DNA sequence containing DrPARG was amplified from the genomic DNA of *D. radiodurans* with the primers DrPARG-F (5′-GGGATCCCCATATGAACCGCAAA-3′, underline contains *Nde*I site) and DrPARG-R (5′-GAATTCTCGAGTCAGGAGGTAGATGAGGGC-3′, underline contains *Xho*I site). The PCR product was inserted between the *Nde*I and *Xho*I restriction sites of the pET28a vector system (Novagen). The expression plasmids of DrPARG mutants were created by site-directed mutagenesis with the primer sets T267R-F (5′-TCAGCACCCGAGGCTCGGC GCGT-3′, underline indicates mutated codons) and T267R-R (5′-ACGCGCC-GAGCCTCGGGTGCTGA-3′, underline indicates mutated codons); T267K-F (5′-TCAGCACCCGAAACTCGGCGCGTTT-3′, underline indicates mutated codons) and T267K-R (5′-AAACGCGCCGAGTTTCGGGTGCTGA-3′, underline sequences indicates mutated codons); E112A-F (5′-GGCGCGCAGGCCCAGGAAGCA GACCTGTGCCGTGGCAGT-3′, underline indicates mutated codons) and E112A-R (5′-ACTGCCACGGCACAGGTCTGCTTCCTGGGCCTGCGCGCC-3′, underline indicates mutated codons). The expression plasmids of wild-type and mutant DrPARG sequences were transformed into *E. coli* BL21 (DE3) cells, which were grown at 37 °C to $OD_{600}$ 1.0 with 50 μg/mL kanamycin. The expression of recombinant protein with a His-tag at the N terminus was induced in cells with 0.5 mM isopropyl-β-D-thiogalactoside, then grown for 20 h at 25 °C. Cells were collected by centrifugation and resuspended in lysis buffer (25 mM phosphate buffer, pH 7.0, 100 mM NaCl). After 20 min of sonication, the cell extract was clarified by centrifugation at $18,900 \times g$ for 30 min at 4 °C to remove debris. The clear supernatant was placed in an open column filled with $Ni^{2+}$-NTA resin. The resin was washed with a 10 bed volume of lysis buffer containing 50 or 100 mM imidazole. The His-tagged DrPARG and its mutants were eluted by lysis buffer containing 300 mM imidazole. The purified protein was further dialyzed against stabilization buffer (25 mM phosphate buffer, pH 7.0, 100 mM NaCl, 0.5 mM dithiothreitol). The His-tag was removed by using thrombin, which resulted in four additional residues (GSHM) at the N terminus. The protein was further purified by gel filtration chromatography with a Superdex75 XK 16/60 column (GE Healthcare) in 20 mM Tris-HCl buffer (pH 7.0) and 100 mM NaCl.

The DNA sequence containing full-length TcPARG was synthesized (MDBio Inc.) and cloned into pET28a vector (Novagen). The expression plasmid of TcPARG sequence was transformed into *E. coli* BL21 (DE3) cells and the expression of recombinant protein was induced in cells with 0.5 mM isopropyl β-D-1-thiogalactopyranoside (IPTG) for 20 h at 16 °C. The N-terminal His-tagged TcPARG was purified by affinity chromatography and eluted by lysis buffer containing 500 mM imidazole. The purified protein was further dialyzed against stabilization buffer (25 mM phosphate buffer, pH 7.0, 100 mM NaCl, 0.5 mM dithiothreitol).

The expression and purification of human PARP1 (HsPARP1) was modified from the previous study[55]. Briefly, the DNA sequence containing full-length HsPARP1 was amplified from pMD-PARP1 (Sino Biological Inc.) and cloned into pET15b vector system (Novagen) using primers HsPARP1-F (5′-AATTCATATGGCGGAGTCTTCGGATAAG-3′, underline contains *Nde*I site) and HsPARP1-R (5′-AATTCTCGAGTTACCACAGGGAGGTCTTAA-3′, underline contains *Xho*I site) and expressed in *E. coli* Rosetta (DE3) cells. The cultured bacterial broth was grown at 37 °C to $OD_{600}$ between 0.4 and 0.6 supplemented with 50 μg/mL ampicillin and 34 μg/mL chloramphenicol, and then added $ZnSO_4$ to a final concentration at 0.1 mM followed by grown at 37 °C. When $OD_{600}$ reached 0.8 to 1.0, the expression of N-terminal His-tagged protein was induced by 0.2 mM IPTG at 16 °C for 3 days. Cells were collected by centrifugation and resuspended in lysis buffer (25 mM HEPES, pH 8.0, 500 mM NaCl, 0.5 mM Tris(2-carboxyethyl)phosphine (TCEP)) supplemented with 0.1% NP-40 and 1 mM phenylmethanesulfonyl fluoride (PMSF). After 40 min of sonication, the cell extract was clarified by centrifugation at $18,900 \times g$ for 30 min at 4 °C to remove debris. The clear supernatant was placed in an open column filled with $Ni^{2+}$-NTA resin (GE Healthcare). The resin was washed sequentially with 10 bed volumes of low-salt wash buffer (25 mM HEPES, pH 8.0, 500 mM NaCl, 0.5 mM TCEP, 20 mM imidazole), 10 bed volumes of high-salt wash buffer (25 mM HEPES, pH 8.0, 1 M NaCl, 0.5 mM TCEP, 20 mM imidazole), and 10 bed volumes of low-salt wash buffer. The recombinant protein was then eluted by elution buffer (25 mM HEPES, pH 8.0, 500 mM NaCl, 0.5 mM TCEP, 400 mM imidazole). The

preliminarily purified protein was diluted by no-salt buffer (50 mM Tris-HCl, pH 7.0, 1 mM EDTA, 0.1 mM TCEP) before applying to a HiTrap Heparin HP Column (GE Healthcare), which was beforehand equilibrated with heparin buffer A (50 mM Tris-HCl, pH 7.0, 1 mM EDTA, 0.1 mM TCEP, and 250 mM NaCl). After applying into the heparin column, the recombinant protein was then eluted by ÄKTA start (GE Healthcare) with a two-step program: wash the column with five bed volumes of heparin buffer A, and elute protein from the column by 25 bed volumes gradient from 0% to 100% of heparin buffer B (50 mM Tris-HCl, pH 7.0, 1 mM EDTA, 0.1 mM TCEP, and 1 M NaCl). The protein was dialyzed against gel filtration buffer (25 mM HEPES, pH 8.0, 150 mM NaCl, 1 mM EDTA, 0.1 mM TCEP) and further purified by gel filtration chromatography with a 10/300 GL Sephacryl$^{TM}$ S-200 HR column (GE Healthcare).

The DNA fragments containing the catalytic domain of human PARP10 (HsPARP10 CatD) (residues 818–1025) were amplified from pCR4-TOPO-hPARP10 (transOMIC technologies) and cloned into pET28a vector system (Novagen) using primers HsPARP10 CatD-F (5′-CGGCATATGAACAACCTGGAGCGTCTGGCA-3′, underline contains NdeI site) and HsPARP10 CatD-R (5′-AATTCTCGAGTTAAGTGTCTGGGGAGCG GCC-3′, underline contains XhoI site) The expression plasmid of HsPARP10 CatD was transformed into E. coli BL21 (DE3) cells and the expression of recombinant protein was induced in cells with 0.1 mM IPTG for 20 h at 25 °C. The N-terminal His-tagged HsPARP10 CatD was purified by affinity chromatography and eluted by lysis buffer (20mM Tris-HCl, pH 8.0, 100 mM NaCl) containing 100 mM imidazole. The protein was further purified by gel filtration chromatography with a Superdex75 XK 16/60 column (GE Healthcare) in buffer containing 20 mM Tris-HCl (pH 8.0), 100 mM NaCl, and 0.5 mM dithiothreitol. DNA sequences of oligonucleotide primers used in cloning for protein expression are listed in Supplementary Table 2.

**CD spectroscopy.** Far-UV CD spectra were measured in 10 μM protein samples with or without pre-incubation with 30 μM ADP-ribose in CD buffer (20 mM phosphate buffer, pH 7.5) placed in a 1 mm pathlength cuvette and recorded on a JASCO J-810 spectropolarimeter equipped with a Peltier temperature control system (JASCO International). Secondary-structure percentage of protein was estimated by using the K2D3 server[56]. Thermal transition of protein samples was monitored at 220 nm from 10 °C to 95 °C at a scan rate of 0.5 °C/min. Baseline subtraction, smoothing, and data normalization involved use of SigmaPlot. The melting temperature ($T_m$) was calculated by interpolation between 0 and 100% unfolded temperature.

**Crystallization and data collection.** Initial crystallization trials were performed at 283 K by the sitting-drop vapor-diffusion method with commercial crystallization screen kits, 96-well Intelli-plates (Art Robbins Instruments) and a Crystal Phoenix robot (Art Robbins Instruments). Each crystallization drop was prepared by mixing 0.3 μL wild-type DrPARG or wild-type/mutant DrPARG mixed with ADP-ribose (molar ratio 1:15) at 10 mg/mL with an equal volume of mother liquor, and the mixture was equilibrated against a 100-μL reservoir solution. The crystals of the wild-type DrPARG apo form were grown at 283 K with an optimal condition of 200 mM lithium sulfate, 100 mM Bis-Tris (pH 6.5), and 25% (v/v) polyethylene glycol 3350. The crystals of the wild-type DrPARG/ADP-ribose complex were grown at 283 K with 100 mM sodium citrate tribasic dihydrate (pH 5.6–6.0), 10–20% (v/v) 2-propanol, or 10–20% (v/v) polyethylene glycol 4000. The crystals of the mutant DrPARG/ADP-ribose complex were grown at 283 K with 100 mM sodium citrate tribasic dihydrate (pH 5.2–5.6), 20% (v/v) 2-propanol, 20% (v/v) polyethylene glycol 4000, and 10 mM hexammine cobalt(III) chloride as the additive. For subsequent diffraction, the crystals were cryoprotected in mother liquor supplemented with 20% glycerol and flash-frozen in liquid nitrogen at 100 K. The diffraction images were recorded in a 100-K nitrogen gas stream with the use of TLS BL15A1, BL13B1, BL13C1, or TPS 05A beamlines (National Synchrotron Radiation Research Center, Taiwan) and processed by using the HKL2000 software[57].

**Structure determination and refinement.** The crystal structure of wild-type DrPARG/ADP-ribose complex was determined by molecular replacement with use of the Phaser-MR program[58] and the PARG structure from T. curvata (PDB code 3SIG)[22] as a search model. The complex structure was further used as the template for determining structures of the wild-type DrPARG apo-form or mutant DrPARG/ADP-ribose complex. Crystallographic refinement involved repeated cycles of conjugate-gradient energy minimization and temperature-factor refinement performed with the program phinex.refine in the PHENIX package[59]. Amino acid side chains and water molecules were fitted into $2F_o$-$F_c$ and $F_o$-$F_c$ electron density maps by using Coot[60]. The model was evaluated with the use of PRO-CHECK[61] and MOLPROBITY[62]. The data collection and structure refinement statistics are in Table 1. Representative electron density maps for the final PARG structures are displayed in stereographic mode in Supplementary Figure 19.

**Bacterial strain and growth conditions.** The bacterial strain wild-type D. radiodurans R1 was purchased from the Food Industry and Development Institute

(Taiwan). Cells were grown in the TGY medium (0.5% tryptone, 0.5% yeast extract, 0.1% glucose, and 0.1% potassium phosphate dibasic) at 30 °C.

**Genomic DNA extraction.** Three milliliters of D. radiodurans R1 ($OD_{600}$ 0.8–1.0) grown in the TGY medium was pelleted by centrifugation at 3500 × g for 5 min. The cells were resuspended with 250 μL TE buffer (50 mM Tris-HCl, pH 8.0, 50 mM EDTA) supplemented with 1 mg/mL lysozyme. The cell lysates were placed on ice for 60 min followed by the addition of 90 μL STEP buffer (0.5% sodium dodecyl sulfate (SDS), 50 mM Tris, pH 8.0, 40 mM EDTA, 2 mg/mL proteinase K) for another 60 min. An amount of 90 μL NH$_4$OAc and 500 μL phenol/chloroform was added to the cell lysates, followed by centrifugation at 6000 × g for 10 min. The aqueous layer was then collected, followed by the addition of 500 μL chloroform/ isoamyl alcohol for centrifugation at 6000 × g for 10 min. The aqueous layer was collected, followed by the addition of 0.6-fold volume of isopropanol and chilling at −80 °C for 30 min. The solution was centrifuged at 800 × g for 5 min, and the supernatant was discarded. The pellet was washed with 70% EtOH two to three times. The pellet was air-dried. The genomic DNA was dissolved in sterile water and stored at −20 °C.

**RNA extraction.** Deinococcus radiodurans R1 ($OD_{600}$ 0.8–1.0) was pelleted by centrifugation at 3500 × g for 5 min. Cells were resuspended in TE buffer (50 mM Tris-HCl, pH 8.0, 50 mM EDTA) with 0.4 mg/ml lysozyme. Total RNA was extracted by using the GeneJET RNA Purification Kit (Thermo Scientific). The RNA quality was examined by 1% agarose gel.

**Detection of endogenous PAR in D. radiodurans.** Three milliliters of cultured D. radiodurans R1/Δparg strain ($OD_{600}$ 0.8–1.0) was pelleted and dissolved in phosphate-buffered saline (PBS) containing 1 mM PMSF (Sigma). Cells were lysed by sonication and the supernatants were separated by centrifugation at 12,000 × g for 10 min. The cell lysates were adjusted to equal protein concentration of 0.6 mg/mL. For dot blot assay, 2 μL lysate was applied onto nitrocellulose membrane (GE Healthcare) and probed with anti-PAR monoclonal antibody, 10H (Cat. No. ALX-804-220-R100; Enzo Life Sciences) (1/10,000) and horseradish peroxidase (HRP)-conjugated anti-mouse IgG (Cat. No. NEF822001EA; PerkinElmer) (1/5000) as primary and secondary antibodies, respectively. The membranes were stained with Ponceau Red (Sigma) as loading controls. For western blot, cell lysates of R1 and Δparg strains were resolved in a 12% acrylamide gel followed by transferring onto a polyvinylidene difluoride (PVDF) membrane (Bio-Rad). The membrane was probed with anti-PAR monoclonal antibody (Enzo Life Sciences) (1/10,000) or anti-pan-ADP-ribose binding reagent (Cat. No. MABE1016; Millipore) (1/5000). For cleavage of endogenous PAR in D. radiodurans, cell lysates of Δparg strain were incubated with or without 5.5 nM human PARG (HsPARG) (Novus Biological) at 37 °C for 1 h. Two microliters of reaction was applied onto nitrocellulose membrane (GE healthcare) and probed with anti-PAR monoclonal antibody, 10H (Enzo Life Sciences) (1/10,000). The membranes of dot blot and western blot were detected for chemiluminescent signals by a FluorChemM imaging system (ProteinSimple).

**Co-immunoprecipitation.** Deinococcus radiodurans R1 ($OD_{600}$ 0.8–1.0) was grown in the TGY medium supplemented with either 25 μM NAD$^+$, biotin (Sigma), or biotinylated NAD$^+$ (Trevigen) at 30 °C for 12–16 h. The culture was pelleted by centrifugation at 6000 × g for 5 min and further lysed by sonication in PBS buffer containing 1 mM PMSF (Sigma). Supernatants were separated by centrifugation at 12,000 × g for 10 min. One microliter of anti-PAR monoclonal antibody, 10H (Enzo Life Science) was added to the supernatant followed by incubation at 4 °C for 2 h. Twenty microliters of Protein A/G-conjugated agarose (Santa Cruz Biotechnology) was added to the supernatants followed by incubation at 4 °C for 2 h. The agarose was washed 2–4 times by PBS and pelleted by centrifugation at 1000 × g for 30 s. The agarose was resuspended in 50 μL PBS containing 1% SDS followed by centrifugation at 12,000 × g for 10 min to detach precipitants. Two microliters of IPed sample was applied onto the nitrocellulose membrane (GE healthcare). The membrane was probed with anti-biotin polyclonal antibody (Cat. No. A150-109A; Bethyl Laboratories Inc.) (1/5000) and HRP-conjugated anti-rabbit IgG (Cat. No. A120-109D2; Bethyl Laboratories Inc.) (1/5000) as primary and secondary antibodies, respectively. Pre-IPed sample was probed with anti-PAR monoclonal antibody (Enzo Life Sciences) (1/10,000) and anti-biotin polyclonal antibody (Bethyl Laboratories Inc.) (1/5000) for detection of cellular contents of PAR and biotin, respectively.

**In vivo PAR synthesis inhibition assay.** Deinococcus radiodurans R1 ($OD_{600}$ 0.8–1.0) was grown in the TGY medium containing various concentrations of 3-aminobenzamide (Sigma) ranging from 100 μM to 2500 μM at 30 °C for 16 h. Cells were pelleted by centrifugation at 6000 × g for 5 min and further lysed by sonication in PBS containing 1 mM PMSF (Sigma). The supernatants were separated by centrifugation at 12,000 × g for 10 min, and the cell lysates were adjusted to equal protein concentration of 0.6 mg/mL. Two microliters of lysate was applied onto the nitrocellulose membrane (GE Healthcare) and probed with anti-PAR monoclonal antibody, 10H (Enzo Life Sciences) (1/10000).

**Generation of *D. radiodurans* Δ*parg* strain**. The gene encoding DrPARG (*parg*) was disrupted by using targeted deletion techniques. Briefly, the DNA fragments containing *parg* were amplified with the primers P1 (5′-CTCTACTCTACGCAG CAGTGATCC-3′) and P2 (5′-GTTCCAGATAGTCGGCGGTGTC-3′) from genomic DNA of *D. radiodurans*. The PCR product was cloned into the TA cloning vector (RBC Bioscience) to create the DrPARG-TA plasmid. The drug cassette conferring kanamycin resistance driven by the *D. radiodurans groEL* promoter was amplified with the primers P3 (5′-ACAGAC<u>AGCGCT</u>TAGAAAAACTCATCGAG CATCAAATG-3′, underline contains the restriction site *Eco*47III) and P4 (5′-TT CTAGG<u>GGGCCC</u>GCCAAGCTCGCGAGGCC-3′, underline contains the restriction site *Apa*I) from the shuttle expression plasmid pRADK (a gift from Dr. Yuejin Hua, Zhejiang University, China)[63]. The PCR product was inserted between the *Eco*47III and *Apa*I restriction sites of the DrPARG-TA plasmid to create the DrPARG-KO plasmid. The DrPARG-KO plasmid was transformed into *D. radiodurans* and recombinants were selected on TGY plates containing 20 μg/mL kanamycin. Disruption was confirmed by PCR and DNA-sequencing. DNA sequences of oligonucleotide primers used in targeted deletion are listed in Supplementary Table 2.

**UV treatment**. An amount of 10 mL cells was placed in Petri dishes ($OD_{600}$ 0.8–1.0) and exposed to UV radiation by using a germicidal UV-C lamp (254 nm) at 0.3 J/m$^2$/s. For growth analysis, irradiated cells were diluted 1:100 in fresh TGY and $OD_{600}$ was measured as a function of time. For detecting PAR level, cells were collected at different times after UV treatment and subjected to dot blot assay with monoclonal anti-PAR antibody.

**Pulsed-field gel electrophoresis**. The PFGE analysis procedure was modified from the previous report[64]. Briefly, irradiated cells were placed in the TGY medium for recovery over a time course. Five hundred microliters of cell suspension was harvested at indicated time points and followed by washing with 0.9% NaCl. The cells resuspended in 0.125 M EDTA (pH 8.0) were equally mixed with 2% CleanCut agarose (Bio-Rad) to a final concentration of 1%. The cell/agarose plugs were lysed by adding lysozyme buffer containing 0.05 M EDTA (pH 7.5) and 1 mg/ mL lysozyme, followed by incubation at 37 °C for overnight. The plugs were further treated with proteinase K buffer containing 0.5 M EDTA (pH 9.0–9.5), 1% lauroyl sarcosine, and 1 mg/mL proteinase K at 50 °C for 6 h, and followed by incubation at 37 °C for 2 days. After treatment of proteinase K, the plugs were washed once with TE buffer (1 mM EDTA, 10 mM Tris-HCl, pH 7.5) supplemented with 1 mM PMSF (Sigma) and washed four times with TE buffer. The genomic DNA containing plugs were further washed with 1× *Not*I Buffer (NEB) for 10–15 min prior to enzyme digestion. Ten units (2 μL/plug) of *Not*I (NEB) was then added into each plug. The reaction mixtures were incubated at 37 °C for overnight. Prior to electrophoresis, the plugs were washed three times with 1× wash buffer (Bio-Rad). The PFGE pattern was resolved on 1% Megabase agarose gels (Bio-Rad) in 0.5% TBE buffer using a CHEF-DRII Pulsed Field Gel Electrophoresis System (Bio-Rad). The electrophoresis was set at 6 V/cm for 18 h at 12 °C with an initial switch time of 10 s and a final switch time of 60 s. Gels were stained with SYBR Gold (Invitrogen) (1/ 10,000) for 30 min and destained in distilled water for 10 min prior to visualization.

**Random-amplified polymorphic DNA analysis**. Irradiated cells were placed in the TGY medium for recovery over a time course. Two milliliters of cell suspension was harvested at indicated time points. The genomic DNA was isolated and the quality was examined by 1% agarose gel. PCR reactions for RAPD analysis were carried out in a total volume of 25 μL. Each reaction mixture contained 200 ng isolated genomic DNA, 0.625 U ExTaq polymerase (Takara), 0.2 mM dNTP mixture (Takara), 2 mM $MgCl_2$, and 1 μM RAPD primer AS-10 (5′-CCCGTCTACC-3′) (MDBio Inc.). The reaction mixtures were amplified for 45 cycles (95 °C for 1 min, followed by 36 °C for 1 min, and 72 °C for 2 min) using a MJ-Mini thermal cycler (Bio-Rad). The PCR products were resolved on a 1% SeaKem LE agarose gel (Lonza) and stained with SYBR Gold (Invitrogen) (1/10,000) prior to visualization. The oligonucleotide primer used in RAPD analysis is listed in Supplementary Table 2.

**MD simulations**. GROMACS software[65] with GROMOS 43a2 force field was used to perform all MD simulations of the PAR trimer in complex with DrPARG. Topology and parameter files for the PAR trimer were obtained by using the PRODRG2 program[66]. The DrPARG-PAR complex model was placed in a periodic box of simple point charge water molecules and neutralized by adding Na$^+$ ions. This complex was subjected to energy minimization (steepest descent, 50,000 steps) and subsequently two phase equilibration (NVT and NPT ensembles) with the protein atoms kept fixed. Then, MD simulations (200 K) were performed for 2 ns with the protein backbone restrained to the X-ray conformation.

**Non-radiometric enzyme kinetic assay**. The protein-free PAR was prepared as described[67,68]. Briefly, HsPARP1 was automodified by incubation of 12 μM purified hPARP1 and 0.7 μM double-stranded DNA (dsDNA) (5′-GGGGTTGCGGC CGCTTGGG-3′) in 20 mM Tris-HCl (pH 7.5), 50 mM NaCl, 5 mM $MgCl_2$, and 0.1 mM TCEP at room temperature for 10 min. NAD$^+$ was then added to the reaction mixture, followed by incubation at room temperature for 30 min. The

reaction was further stopped by 0.625 mM 3-aminobenzamide. To release PAR from automodified HsPARP1, releasing buffer containing 100 mM CHES (2-(cyclohexylamino)ethanesulfonic acid) (pH 9.0) and 10 mM EDTA was added into reaction. The reaction mixture was stood at room temperature for 1 h. Equal volume of phenol/chloroform was added to extract PAR polymers followed by vortexing for 10 min. The mixture was then pelleted by centrifugation at 9600 × *g* for 10 min. The aqueous layer was transferred to a new tube, followed by the addition of 1/10 volume of 3 M NaOAc (pH 4.0) and 2 volumes of ethanol. The reaction was kept at −80 °C for 30 min, followed by centrifugation at 16,200 × *g* for 15 min. After discarding the supernatant, the pellet was washed twice with 70% ethanol and air-dried. Dried pellet was dissolved in storage buffer containing 10 mM Tris-HCl (pH 8.0), 10 mM EDTA.

The purified PAR was measured at $A_{258}$ by using NanoDrop 1000 (Thermo Fisher Scientific) and the absorbance value was converted to molar concentration with the molar absorptivity constant of PAR at 258 nm (13,500 cm/M). Enzyme kinetics of wild-type and mutant DrPARG were analyzed by an antibody-based quantitative dot blot assay system[69]. Briefly, 2 μL purified acceptor-free PAR (0.62–4 μM) in reaction buffer (20 mM phosphate, pH 7.0, 100 mM NaCl) were applied onto nitrocellulose membrane (GE Healthcare) and probed with anti-PAR monoclonal antibody (Enzo Life Sciences) (1/10,000) to generate calibration curves. Wild-type or mutant DrPARG (60 ng/mL) was incubated with PAR (0.08–2.5 μM) at 37 °C for 20 min. The reactions were terminated by adding 1% SDS, and 2 μL reaction mixture was applied onto nitrocellulose membrane (GE healthcare) and probed with anti-PAR monoclonal antibody (Enzo Life Sciences) (1/10,000). The chemiluminescent images were analyzed using ImageJ 1.49v (NIH). Data fitting of Michaelis–Menten plot and kinetic parameter calculations involved use of Prism 7.0a (GraphPad software).

**PARG cleavage assay**. One micromole of wild-type or mutant DrPARG was added to the reaction buffer (20 mM Tris-HCl, pH 7.5, 50 mM NaCl, 5 mM $MgCl_2$, and 0.1 mM TCEP) containing 1 μM automodified HsPARP1. The reactions were incubated at 37 °C and aliquots were taken at 1, 5, 10, 15, 30, or 60 min and terminated by adding 1% SDS. Two microliters of reaction was applied onto the nitrocellulose membrane (GE healthcare) and probed with anti-PAR monoclonal antibody (Enzo Life Sciences) (1/10,000). For detecting endo-glycohydrolase activity, the reactions treated with DrPARG (wild-type or mutants), TcPARG, or HsPARG (Novus Biologicals) were applied into a 30 kDa cut-off Nanosep centrifugal device (Pall) at indicated time points and centrifuged at 5000 × *g* for 5 min. Ten microliters of filtrate was applied onto nitrocellulose membrane (GE healthcare) and probed with anti-PAR monoclonal antibody (Enzo Life Sciences) (1/ 10,000). For cleavage of endogenous PAR in *D. radiodurans*, cell lysates of Δ*parg* strain were incubated with or without 600 nM WT/E112A DrPARG at 30 °C. Aliquots of reaction were taken at indicated time points and terminated by adding 1% SDS. Two microliters of reaction was applied onto the nitrocellulose membrane (GE healthcare) and probed with anti-PAR monoclonal antibody (Enzo Life Sciences) (1/10,000).

**High-performance liquid chromatography**. Automodified HsPARP1 was incubated with or without WT/mutant DrPARG for 15 min and the reactions were applied into 10 kDa cut-off Nanosep centrifugal devices (Pall) followed by centrifugation at 5000 × *g* for 5 min. The analysis of the filtrates was performed using a reverse-phase L-7100 HPLC system (Hitachi) coupled with a 250 × 4.6 mm$^2$ $C_{18}$ HPLC column (Phenomenex). The HPLC elution procedure was modified from previous report[28]. The A eluent containing 5 mM pentylamine adjusted to pH 6.5 by titration of acetic acid and the B eluent containing 100% acetonitrile were used for elution. The flow rate was set to 1 mL/min. The gradient elution program was starting with 2% B eluent, and then the percentage of B eluent was linearly increased to 25% in 5 min, followed by an isocratic elution of 25% B eluent for another 10 min. The peak elution products were collected manually for further MS analysis.

**ESI-Q-TOF mass spectrometry**. Before ESI-Q-TOF MS (electrospray ionization-quadrupole-time-of-flight MS) analysis, the sample was separated by HPLC for desalting and pre-concentration as described above. For ESI-Q-TOF analysis, the ADP-ribose oligomers were identified by the ESI connected to a hybrid Q-TOF mass spectrometer (Waters, Millford, MA, USA). The SYNAPT HDMS G1 was operated in negative ionization mode with electrospray capillary voltage at 1.2 kV. All mass spectra were recorded in the continuous scan mode for negative ions with the scan range of *m*/*z* from 800 to 1200.

**De-mono glutamate-linked ADP-ribosylation assay**. To perform deMARylation assay, reaction mixture containing 10 μM HsPARP10 CatD and 100 μM biotinylated NAD$^+$ (Trevigen) in reaction buffer (20 mM Tris-HCl, pH 8.0, 100 mM NaCl, and 0.5 mM dithiothreitol) was incubated at room temperature for 30 min. The reaction mixture was dialyzed against reaction buffer to remove residual NAD$^+$. Five micromoles of yeast Poa1p (ScPoa1p) or various concentrations of DrPARG (2.5–10 μM) was added into the reaction, followed by incubation at room temperature for another 30 min. The reaction was terminated by adding 1% SDS and proteins were resolved in 15% acrylamide gel. The gel was transferred onto a

PVDF membrane (Bio-Rad) and probed with anti-biotin polyclonal antibody (Bethyl Laboratories Inc.) (1/5000).

**De-mono serine-linked ADP-ribosylation assay**. Reaction mixture containing 0.5 μM HsPARP1, 0.5 μM human HPF1 (CUSABIO), 1 μM activated dsDNA (5′-GGGTTGCGGCCGCTTGGG-3′), 50 μM biotinylated NAD+ (Trevigen), and 30 μg human histone H3 peptide (a.a. 1–21) (Sigma) in reaction buffer (50 mM Tris-HCl, pH 8.0, 100 mM NaCl, and 2 mM MgCl$_2$) was incubated at room temperature for 30 min to generate serine-linked mono ADP-ribosylated H3 peptide. The reaction was terminated by adding 3 μM PARP1 inhibitor, BYK04365 (Sigma). One micromole of human ARH3 (HsARH3) (Enzo Life Sciences) or various concentrations of DrPARG (1–3 μM) was added into the reaction, followed by incubation at 37 °C for another 30 min. The reaction was terminated by adding 1% SDS and proteins were resolved in 20% acrylamide gel. The gel was transferred onto a PVDF membrane (Bio-Rad) and probed with anti-biotin polyclonal antibody (Bethyl Laboratories Inc.) (1/5000).

**Isothermal titration calorimetry**. Binding of ADP-ribose to wild-type or mutant DrPARG was measured by ITC with the Nano Isothermal Titration Calorimeter (TA Instruments). Aliquots of 4 μL of 1.5–3 mM ADP-ribose were injected into sample cells containing 0.075–0.15 mM protein sample in 20 mM Tris-HCl, pH 7.0, and 100 mM NaCl. ITC experiments were executed at 25 °C with 250-r.p.m. stirring speed. Additional background heat from titration of ADP-ribose to buffer was subtracted during data analysis. The corrected heat was used for deriving parameters of stoichiometry of the binding ($n$), association constant ($K_a$), dissociation constant ($K_d$), apparent enthalpy of binding ($\Delta H$), and entropy change ($\Delta S$). Data were fitted by applying an independent binding model with NanoAnalyze v2.3.6.

**Statistical analysis**. Data are represented as mean ± standard error of the mean (SEM). $P$ value was determined by Student's $t$ test with the use of Prism 7.0a (GraphPad software).

**Reporting summary**. Further information on experimental design is available in the Nature Research Reporting Summary linked to this article.

## Data availability

The crystallographic data that support the findings of this study are available in the PDB with the identifiers: 5ZDA for apo DrPARG ($P2_12_12_1$); 5ZDB for ADP-ribose-bound DrPARG ($P2_1$); 5ZDC for ADP-ribose-bound DrPARG ($P3_2$); 5ZDD for ADP-ribose-bound DrPARG ($P2_12_12_1$); 5ZDE for ADP-ribose-bound DrPARG ($P3_221$); 5ZDF for ADP-ribose-bound T267K mutant ($P3_221$); and 5ZDG for ADP-ribose-bound T267R mutant ($P3_221$). The source data underlying Fig. 3, Figs. 5a–e, and Supplementary Figs. 1A, 7B, 8, 9, 10, 11, 13, and 15 are provided as a Source Data file. A reporting summary for this Article is available as a Supplementary Information file. All other data are available from the corresponding author upon reasonable request.

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

## Acknowledgements

We thank Dr. Yuejin Hua (Zhejiang University, China) for providing the shuttle plasmid pRADK. We also thank the Technology Commons in College of Life Science and Center for Systems Biology, National Taiwan University for instrumental support of protein crystallization. This work was supported by the Ministry of Science and Technology, Taiwan (MOST 107-2113-M-002-013, MOST 106-2113-M-002-020, MOST 105-2113-M-002-009 and MOST 103-2113-M-002-009-MY2) and National Taiwan University (NTU-CDP-108L7861, NTU-CDP-106R7867, NTU-ERP-104R8600). Portions of this research were carried out at the National Synchrotron Radiation Research Center, a national user facility supported by the National Science Council of Taiwan. The Synchrotron Radiation Protein Crystallography Facility is supported by the National Core Facility Program for Biotechnology. We thank Laura Smales for copyediting the manuscript.

## Author contributions

C.-H.H. conceived and supervised the study. C.-C.C., C.-Y.C., Y.-C.C., and M.-H.L. performed the biochemical and biophysical experiments. C.-C.C., C.-Y.C., and M.-H.L. conducted X-ray protein crystallography. C.-C.C., C.-Y.C., and Y.-C.C. performed cell experiments. C.-C.C., C.-Y.C., and C.-H.H. performed bioinformatics and structural analyses. C.-C.C., C.-Y.C., and C.-H.H. wrote the paper. All authors approved the final manuscript.

## Additional information

**Competing interests:** The authors declare no competing interests.

