## [Peer Review File · Nature Communications]

Reviewers' comments:

Reviewer #1 (Remarks to the Author):

The manuscript authored by Chao-Cheng Cho et al. and entitled “Structural and biochemical evidence supporting poly ADP-ribosylation in the bacterium *Deinococcus radiodurans*” reports the structural and biochemical characterization of the PARG enzyme from the extreme radiation-resistant organism, *D. radiodurans*, and provides substantial experimental evidence for the existence of a functional PARylation machinery in this bacterium. This is the first report of such activity in prokaryotes and represents a major finding.

Although the overall work is clearly presented and the structural studies are sound, this study would benefit from additional experiments and analyses, and care should be taken to avoid overstatements. The major points are listed below.

1. An in-depth structural comparison of the DrPARG structure with the eukaryotic homologues should be performed as well as a comparative study with TcPARG, since canonical PARGs possess both exo- and endo-activity and would thus be more similar to DrPARG. How different are their active sites?
2. Related to point 1, a sequence alignment of PARGs should be provided including the sequences discussed in the manuscript (TcPARG, DrPARG, human PARG etc.). How conserved are Thr267 and Glu112? Do most bacterial species have an Arg like TcPARG or a Thr like DrPARG?
3. The quality of dot blots is insufficient. Ponceau staining reveals different amounts were deposited (see also minor point 1 below) and the data presented in Fig. 3A and in Fig.3E for WT DrPARG are quite different. How could such data be used for a quantitative analysis? How much cell lysate was used in Fig.3A? It would be better to deposit a defined amount of protein (after protein dosage of the cell lysate).
4. Examples of overstatements: ‘PARG might be involved in DNA damage repair in response to radiation damage’ – many other processes are upregulated following radiation damage, so this is no evidence for a role in DNA repair. ‘endogenous PAR is mainly regulated by DrPARG’ - no information about other types of regulation and the importance of these potential regulators... ‘decreased binding affinity of PAR and further implies the loss of endo-glycohydrolase activity’ – not necessarily true. Loss of end-glycohydrolase activity is supported by absence of short chain PAR in the filtrates.
5. Potential role of DrPARG in DNA damage repair should be evaluated. For example, by running pulse-field gel electrophoresis experiments on genomic DNA before and after irradiation in WT and Δ PARG strains to compare kinetics of repair.

6. PARG in vitro cleavage assay (Fig. S9 and S11). Error in legend: t=0 should be without PARG and not without PARP1. Why does short-chain PAR disappear after 15 min timepoint? Are small PAR fragments no longer detected by the antibody? Would be nice to have a positive (canonical PARG with endo- and exo activity) and a negative (TcPARG) in this assay for comparison. This data seems critical for the conclusions of the manuscript and should be presented in the main text.
7. No data to support the statement 'The detected short-chain PAR did not likely result from removal of short PAR from PARP1 but rather from endo-cleavage of O-glycosidic linkages of long-chain PAR because PARG could not hydrolyze the ester bond attached to protein'.
8. ITC data: what binding model was used for fitting the data? What is the evidence suggesting that mutation of Thr267 'impairs the ability of binding to the internal ADP-ribose units (Table 2)?
9. Based on the findings presented in this paper, are most prokaryotic PARG predicted to be exo-glycohydrolases or endo- and exo-glycohydrolases? This question should be addressed by comparing PARG sequences. Such a discussion would also benefit from comparisons with eukaryotic canonical PARG enzymes.

Additional minor points:

1. More details should be provided to describe the dot blot assay: how much material was deposited or culture volume used? Which secondary antibody? Which mode of detection (radiolabel, colorimetric etc.)?
2. Page 11: Co-IP with PAR antibody confirmed co-localization. Do the authors mean 'interaction'? The following statement is also unclear: '...thus suggesting that the PAR signal was occasioned from NAD+'. Do you mean, PAR signal is dependent on the availability of NAD+?
3. Last line of page 16, '...suggested that the exo-glycohydrolase activity was blocked'. This should be the 'endo-glycohydrolase activity'. Also on page 20, line 375: 'Our data suggest the presence of exo-glycosidic cleavage of PAR in prokaryotic microorganisms.' It should be both endo- and exo- activities.
4. Description of structural data regarding mutant DrPARG in Fig. 6 is unclear.
5. In the results section, it is not clearly stated that MD simulations were performed with a tri-ADP-ribose, but only that a model was created. This is not the same and should be clarified.

Reviewer #2 (Remarks to the Author):

Cho and colleagues solved the crystal structure of ADP-ribose-bound bacterial type PARG from *Deinococcus radiodurans* (DR). The authors further suggested that the DR PARG possesses both exo- and endoglycohydrolase activities (in contrast to the previously characterised bacterial PARG from *Thermomonospora curvata*). Finally, the authors constructed the PARG-deficient *Deinococcus* strain and provided the first in vivo evidence of endogenous poly(ADP-ribose) (PAR) metabolism in any bacterial organism. This paper is timely and well-written; most of the conclusions in the paper are supported by the results.

Major comments:

Figure 3. Please provide one panel where the PAR in the cell lysate is analysed by the WB rather than dot blot. This would also give an idea of the size of the PARylated proteins. It would be important to try both the PAR antibody as well as the anti-pan-ADP-ribose binding reagent (Millipore).

Figure 3D. It would be ideal to see here an alternative/more direct evidence that *Deinococcus* cell is permeable for biotinylated NAD.

Figure 3E. Please provide an additional control here: treatment of the cell lysate with the recombinant human PARG enzyme.

Figure 5 Claim that the DR PARG possesses endo-glycohydrolase activity should be supported by a direct experimental evidence (for example by LC-MS or sequencing gels). Also, it is quite possible that the DR PARG is capable of cleaving the link between PAR chain and protein. To test this I would suggest preparing the glutamate- and serine-linked mono(ADP-ribosyl)ated protein substrates (for example as described in Fontana et al, *Elife*, 2017) and see whether DR PARG can de-modify these substrates.

Does the growth of *Deinococcus* in the presence of 3-aminobenzamide block cellular PAR signal (as demonstrated for PAR formation in *Streptomyces* species)?

Minor comments:

Page 3 line 50. Please also add here the original reference on the discovery of PAR-binding zinc-fingers (Ahel et al, Nature, 2008).

There are several typos throughout the text. For example, on page 18 line 324 - 'RARylation'

Point-by-point response to Reviewers' comments

-- Reviewer comments in italics, our answers in regular font --

We thank the Reviewers and Editor for their valuable comments. We have addressed the points in the revised manuscript. All corrections made in the revised manuscript/figure legends are marked in red. Please see the point-by-point response as follows:

Reviewers' comments:

Reviewer #1 (Remarks to the Author):

The manuscript authored by Chao-Cheng Cho et al. and entitled “Structural and biochemical evidence supporting poly ADP-ribosylation in the bacterium *Deinococcus radiodurans*” reports the structural and biochemical characterization of the PARG enzyme from the extreme radiation-resistant organism, *D. radiodurans*, and provides substantial experimental evidence for the existence of a functional PARylation machinery in this bacterium. This is the first report of such activity in prokaryotes and represents a major finding.

Although the overall work is clearly presented and the structural studies are sound, this study would benefit from additional experiments and analyses, and care should be taken to avoid overstatements. The major points are listed below.

1. An in-depth structural comparison of the DrPARG structure with the eukaryotic homologues should be performed as well as a comparative study with TcPARG, since canonical PARGs possess both exo- and endo-activity and would thus be more similar to DrPARG. How different are their active sites?

We are very grateful to the reviewer for this point. A detailed comparison of the ADP-ribose binding pocket in DrPARG, TcPARG, and HsPARG was performed as suggested and re-drawn Fig. 2 and Supplementary Fig. S6 for the comparison. The results showed that both canonical and bacterial PARGs adopted similar organization of amino acids interacting with ADP-ribose in the active sites, as well as their catalytic mechanism. However, structural features such as the ribose cap (only appears in bacterial PARG) block the extension of n+1 ADP-ribose moiety and

distinguish bacterial PARG from canonical PARG. DrPARG also has the ribose cap, however, it can act as endo-PARG because Thr267 does not confer steric hinderance.

2. Related to point 1, a sequence alignment of PARGs should be provided including the sequences discussed in the manuscript (TcPARG, DrPARG, human PARG etc.). How conserved are Thr267 and Glu112? Do most bacterial species have an Arg like TcPARG or a Thr like DrPARG?

Thanks for the comments. A structure-based sequence alignment was performed and shown in Supplementary Fig. S6. Glu112 is the conserved catalytic residue among bacterial and canonical PARG. Thr267 is not conserved among bacterial and canonical PARG because it resides in the ribose cap structure which only found in bacterial PARG. When we performed BLAST against DrPARG sequences, potential bacterial PARGs can be found. We aligned the sequences of the proteins and found the occurrence of Thr is higher than Arg in the position corresponding to Thr267. Many of the proteins have Thr or Val in the position corresponding to Thr267.

3. The quality of dot blots is insufficient. Ponceau staining reveals different amounts were deposited (see also minor point 1 below) and the data presented in Fig. 3A and in Fig.3E for WT DrPARG are quite different. How could such data be used for a quantitative analysis? How much cell lysate was used in Fig.3A? It would be better to deposit a defined amount of protein (after protein dosage of the cell lysate).

We are very grateful to the reviewer for this point. The assay was re-done by quantifying the total protein in the lysates. 3mL culture were taken for preparing the lysate and total protein was quantified using Bradford assay before dot blotting (Fig. 3E).

4. Examples of overstatements: 'PARG might be involved in DNA damage repair in response to radiation damage' – many other processes are upregulated following radiation damage, so this is no evidence for a role in DNA repair. 'endogenous PAR is mainly regulated by DrPARG' - no information about other types of regulation and the importance of these potential regulators... 'decreased binding affinity of PAR and further implies the loss of endo-glycohydrolase activity' – not necessarily true. Loss of end-glycohydrolase activity is supported by absence of short chain PAR in the filtrates.

We are very grateful to the reviewer for this point. We modified the sentences as “endogenous PAR is regulated by DrPARG..” and “which indicates decreased binding affinity of PAR..” to avoid overstatements. In addition, more experiments such as PFGE (Pulsed Field Gel Electrophoresis) and RAPD (Random-Amplified Polymorphic DNA) were performed to examine the role of DrPARG in DNA damage repair and the results suggested DrPARG is involved in DNA repair.

5. Potential role of DrPARG in DNA damage repair should be evaluated. For example, by running pulse-field gel electrophoresis experiments on genomic DNA before and after irradiation in WT and Δ PARG strains to compare kinetics of repair.

We are very grateful to the reviewer for this point. As reviewer’s suggestion, Pulsed Field Gel Electrophoresis (PFGE) experiment was performed examining the role of DrPARG in DNA damage repair and the results showed disruption of DrPARG compromises recovery of genome after radiation damage (Supplementary Fig. S9), suggesting DrPARG is involved in DNA damage repair. In addition, another experiment RAPD (Random-Amplified Polymorphic DNA) for evaluation of DNA-damage repair was performed and the results also indicated DrPARG is involved in DNA repair (Supplementary Fig. S10).

6. PARG in vitro cleavage assay (Fig. S9 and S11). Error in legend: $t=0$ should be without PARG and not without PARP1. Why does short-chain PAR disappear after 15 min timepoint? Are small PAR fragments no longer detected by the antibody? Would be nice to have a positive (canonical PARG with endo- and exo activity) and a negative (TcPARG) in this assay for comparison. This data seems critical for the conclusions of the manuscript and should be presented in the main text.

Thanks for the reviewer’s kind reminding. We corrected the figure legend and added positive (HsPARG) and negative (TcPARG) controls as suggested.

The PAR monoclonal antibody 10H preferentially recognize PAR >20 mer. After 15 min reaction, short-chain PAR may be processed to smaller fragments beyond the detection limit of antibody and we added the description in the result section of main text. We also added the controls of TcPARG and HsPARG to the assay(Fig. 5B).

7. No data to support the statement ‘The detected short-chain PAR did not likely result from removal of short PAR from PARP1 but rather from endo-cleavage of

O-glycosidic linkages of long-chain PAR because PARG could not hydrolyze the ester bond attached to protein'.

We are very grateful to the reviewer for this point. Thus, we conducted de-MARylation experiments on DrPARG as shown in Fig. S15. The results suggested DrPARG is incapable of removing glutamate- or serine-linked mono ADP-ribosylation. We also modified the sentence in the main text.

8. ITC data: what binding model was used for fitting the data? What is the evidence suggesting that mutation of Thr267 'impairs the ability of binding to the internal ADP-ribose units (Table 2)?

The ITC data were fitted using an independent binding model. The mutation of Thr267 may impair the binding of internal ADP-ribose unit was inferred from the increased K_m of T267 mutant in Table 2 not the ITC data. We deleted the sentence to avoid misunderstanding.

9. Based on the findings presented in this paper, are most prokaryotic PARG predicted to be exo-glycohydrolases or endo- and exo-glycohydrolases? This question should be addressed by comparing PARG sequences. Such a discussion would also benefit from comparisons with eukaryotic canonical PARG enzymes.

We are very grateful to the reviewer for this point. We performed BLAST against the sequence of DrPARG and many microbial proteins bearing the PARG signature motif were identified, suggesting they may be potential PARGs. We aligned the sequences of those proteins and found many of them have threonine and valine in the position corresponding to Thr267, suggesting they may be endo-PARGs. However, the number of bacterial endo-PARG may be underestimated since Thr267 is not the only determinant of endo-glycohydrolase activity. The endo-activity may be better inferred when structural flexibility is taken considered.

Additional minor points:

1. More details should be provided to describe the dot blot assay: how much material was deposited or culture volume used? Which secondary antibody? Which mode of detection (radiolabel, colorimetric etc.)?

Thanks for the reviewer's kind reminding. We provided more details regarding the culture volume and the secondary antibody used, as well as the mode of detection to

describe the dot blot assay in Methods **“Detection of Endogenous PAR in *D. radiodurans* by Dot Blot Assay and Western Blotting”**.

2. Page 11: Co-IP with PAR antibody confirmed co-localization. Do the authors mean ‘interaction’? The following statement is also unclear: ‘..thus suggesting that the PAR signal was occasioned from NAD⁺’. Do you mean, PAR signal is dependent on the availability of NAD⁺?

Thanks for the reviewer’s kind reminding.

Co-IP experiment was used for examining the incorporation of labelled-NAD⁺ into PAR and verifying the existence of endogenous PAR since NAD⁺ is the substrate for generating PAR. We think environmental abundance of NAD⁺ may affect endogenous PAR level because when we supplemented NAD⁺ to bacterial culture, PAR signal increased (Fig. 5D). We corrected the statement to “...suggesting that the PAR signal was dependent on the availability of NAD⁺”.

3. Last line of page 16, ‘...suggested that the *exo*-glycohydrolase activity was blocked’. This should be the ‘*endo*-glycohydrolase activity’. Also on page 20, line 375: ‘Our data suggest the presence of *exo*-glycosidic cleavage of PAR in prokaryotic microorganisms.’ It should be both *endo*- and *exo*- activities.

Thanks for the reviewer’s kind reminding. We modified the sentences to “...which suggested that the *endo*-glycohydrolase activity was impaired” and “suggest the presence of both *exo*- and *endo*-activities of PARG in prokaryotic microorganisms.” as suggested.

4. Description of structural data regarding mutant DrPARG in Fig. 6 is unclear.

We added additional descriptions regarding the space groups of T267R and T267K structures and their RMSD to WT structure, as well as the differences in solvent accessible surface of ADP-ribose binding site in WT, T267R, and T2677K structures (Fig. S17).

5. In the results section, it is not clearly stated that MD simulations were performed with a tri-ADP-ribose, but only that a model was created. This is not the same and should be clarified.

Thank you for your kind reminding. MD simulation was for minimization and

refinement of the complex model. We modified the sentence to “we constructed the structural model of DrPARG in complex with tri-ADP-ribose using MD simulation”

Reviewer #2 (Remarks to the Author):

Cho and colleagues solved the crystal structure of ADP-ribose-bound bacterial type PARG from *Deinococcus radiodurans* (DR). The authors further suggested that the DR PARG possesses both exo- and endoglycohydrolase activities (in contrast to the previously characterised bacterial PARG from *Thermomonospora curvata*). Finally, the authors constructed the PARG-deficient *Deinococcus* strain and provided the first in vivo evidence of endogenous poly(ADP-ribose) (PAR) metabolism in any bacterial organism. This paper is timely and well-written; most of the conclusions in the paper are supported by the results.

Major comments:

Figure 3. Please provide one panel where the PAR in the cell lysate is analysed by the WB rather than dot blot. This would also give an idea of the size of the PARylated proteins. It would be important to try both the PAR antibody as well as the anti-pan-ADP-ribose binding reagent (Millipore).

We are very grateful to the reviewer for this point. As suggestion from reviewer, we provided a panel for the PAR in the cell lysate analyzed by the WB using both PAR antibody and the anti-pan-ADP-ribose binding reagent (Millipore) shown in Fig. 3C. Methods and related statement were described in main text.

*Figure 3D. It would be ideal to see here an alternative/more direct evidence that *Deinococcus* cell is permeable for biotinylated NAD.*

Thanks for reviewer's comments. To provide the evidence, we added a control treating the culture with biotin (Fig. 3E). In the preIP sample analyzed by dot blot using biotin antibody. The biotin signal in cells treated with biotin-NAD⁺ or biotin, was stronger than that treated with non-labelled NAD⁺, suggesting *Deinococcus* cell is permeable for biotin-NAD⁺.

Figure 3E. Please provide an additional control here: treatment of the cell lysate with the recombinant human PARG enzyme.

Thanks for reviewer's suggestion. We provided a dot blot for treating the cell lysates of Δ PARG cell with recombinant HsPARG (Fig. 3H).

Figure 5 Claim that the DR PARG possesses endo-glycohydrolase activity should be supported by a direct experimental evidence (for example by LC-MS or sequencing gels). Also, it is quite possible that the DR PARG is capable of cleaving the link between PAR chain and protein. To test this I would suggest preparing the glutamate- and serine-linked mono(ADP-ribosyl)ated protein substrates (for example as described in Fontana et al, Elife, 2017) and see whether DR PARG can de-modify these substrates.

We are very grateful to the reviewer for this point. As suggestion from reviewer, we conducted the follow experiments. The cleavage products of DrPARG was separated by HPLC and analyzed by Q-TOF MS. Oligo ADP-ribose such as dimer, trimer and tetramer were identified. De-MARylation experiments were performed with DrPARG using PARP10 catalytic domain and human H3 peptide (a.a. 1-21) as glutamate- and serine-linked mono ADP-ribosylated protein substrates, respectively (Fig. S15). The results suggested DrPARG is incapable of cleaving the glutamate- and serine-linked mono ADP-ribosylation. Figures and statements of the results as well as experimental methods were added into the main text.

Does the growth of Deinococcus in the presence of 3-aminobenzamide block cellular PAR signal (as demonstrated for PAR formation in Streptomyces species)?

We are very grateful to the reviewer for this point. Thus, we treated the R1 cells with various amounts of 3-aminobenzamide for overnight culture. The result of dot blot suggested that 3-aminobenzamide can block cellular PAR signal as shown in Fig. 3F.

Minor comments:

Page 3 lane 50. Please also add here the original reference on the discovery of PAR-binding zinc-fingers (Ahel et al, Nature, 2008).

Thanks for the kind reminding. We added the reference in the text.

There are several typos throughout the text. For example, on page 18 line 324 - 'RARylation'

Thanks for the kind reminding. We corrected the typos and checked others.

Reviewers' comments:

Reviewer #1 (Remarks to the Author):

The authors have addressed all the issues raised in my initial review and have substantially improved their manuscript with the inclusion of a large amount of new data. I thus strongly recommend that this manuscript be published in Nature Communications.

Reviewer #2 (Remarks to the Author):

The authors have addressed most of my concerns. However, the authors tested two ADP-ribosylation antibodies against the wild type extract by western, but, surprisingly, not against the PARG-deficient extract (new figure 3C). Since the two antibodies used show very different profiles, it would be particularly important to confirm that at least some of the protein bands are affected by the PARG status.

Point-by-point response to Reviewers' comments

-- Reviewer comments in italics, our answers in regular font --

We thank the Reviewers and Editor for their valuable comments. We have addressed the points in the revised manuscript. All corrections made in the revised manuscript/figure legends are marked in red. Please see the point-by-point response as follows:

Reviewers' comments:

Reviewer #1 (Remarks to the Author):

The authors have addressed all the issues raised in my initial review and have substantially improved their manuscript with the inclusion of a large amount of new data. I thus strongly recommend that this manuscript be published in Nature Communications.

Ans:

We are thankful for this appreciation of our manuscript.

Reviewer #2 (Remarks to the Author):

The authors have addressed most of my concerns. However, the authors tested two ADP-ribosylation antibodies against the wild type extract by western, but, surprisingly, not against the PARG-deficient extract (new figure 3C). Since the two antibodies used show very different profiles, it would be particularly important to confirm that at least some of the protein bands are affected by the PARG status.

Ans:

We thank you for your valuable suggestions. We performed western blots using PAR antibody and pan-ADP-ribose binding reagent against both wild-type R1 and PARG-deficient cell lysates (Fig. 3F). The results showed some protein bands exhibited higher intensities in Δ PARG cells compared with those in R1 cells detected by both PAR antibody and ADP-ribose-binding reagent, suggesting the modification of protein was affected by PARG status. We have conducted a new figure 3 and included the description of western-blot results in the text.